**COMMUNICATIONS**

# A distal centriolar protein network controls organelle maturation and asymmetry

Lei Wang[1], Marion Failler[1], Wenxiang Fu[1,2] & Brian D. Dynlacht[1]

A long-standing mystery in the centrosome field pertains to the origin of asymmetry within the organelle. The removal of daughter centriole-specific/enriched proteins (DCPs) and acquisition of distal appendages on the future mother centriole are two important steps in the generation of asymmetry. We find that DCPs are recruited sequentially, and their removal is abolished in cells lacking *Talpid3* or *C2CD3*. We show that removal of certain DCPs constitutes another level of control for distal appendage (DA) assembly. Remarkably, we also find that Talpid3 forms a distal centriolar multi-functional hub that coordinates the removal of specific DCPs, DA assembly, and recruitment of ciliary vesicles through distinct regions mutated in ciliopathies. Finally, we show that Talpid3, C2CD3, and OFD1 differentially regulate the assembly of sub-distal appendages, the CEP350/FOP/CEP19 module, centriolar satellites, and actin networks. Our work extends the spatial and functional understanding of proteins that control organelle maturation and asymmetry, ciliogenesis, and human disease.

[1] Department of Pathology, New York University Cancer Institute, New York University School of Medicine, New York, NY 10016, USA. [2]Present address: Biozentrum, University of Basel, 4056 Basel, Switzerland. Correspondence and requests for materials should be addressed to B.D.D. (email: brian.dynlacht@nyumc.org)

The centrosome is an asymmetric organelle comprised of a daughter centriole, a mother centriole, pericentriolar material, and pericentriolar satellites. The older, mother centriole is distinguished by distal appendages (DA) and sub-distal appendages (SDA), whereas the younger centriole is characterized by daughter centriole-specific/enriched proteins (DCPs). DCPs, including CEP120, Centrobin, and Neurl4, are recruited to nascent daughter centrioles after centriole duplication is initiated to regulate centriole elongation and homeostasis, and they are subsequently removed at the G1/S transition in the next cell cycle during mother centriole maturation[1–6]. Subsequently, SDA and DA are assembled at the maturing mother centriole during the G2 phase and the G2/M transition, respectively. DA, assembled through the sequential recruitment of CEP83, CEP89, SCLT1, CEP164, and FBF1 proteins, are involved in vesicle docking and intraflagellar transport (IFT) during ciliogenesis and in immune synapse formation[7–13]. SDA, composed of ODF2, Centriolin, CEP128, CEP170, and other proteins, are required for microtubule anchoring and cilium positioning[14–18]. Mutations in genes linked to mother centriole maturation are associated with a plethora of human diseases, termed ciliopathies, including Joubert syndrome (JBTS), Jeune asphyxiating thoracic dystrophy, Bardet–Biedl syndrome, and oral–facial–digital (OFD) syndrome, among others[19–23].

Despite the above observations, the regulatory mechanisms linking the recruitment of SDA and DA and other distal-end proteins to mother centriole maturation are largely unknown. To our knowledge, only three proteins (C2CD3, OFD1, and ODF2) have been linked to mother centriole maturation. Intriguingly, however, C2CD3 and OFD1 localize on centriolar satellites (CS) and the distal ends of both mother and daughter centrioles, where they play antagonistic roles in centriole elongation. C2CD3 is involved in DA, and perhaps SDA, assembly[24,25], whereas its interacting partner, OFD1, appears to be required for DA assembly only[26]. ODF2, a SDA component, initiates appendage assembly through direct interactions with other SDA proteins, but its role in DA assembly, if any, remains to be clarified[14,16]. Further, although the constellation of DA and SDA proteins have been identified and localized with ever-increasing precision, the relationship between the assembly of these structures, removal of DCPs, and mother centriole maturation—the basis of asymmetry within the organelle—remains largely uncharacterized. To this end, it is attractive to speculate that mother and daughter centriole components display antagonistic relationships to suppress or promote their respective identities.

Talpid3 is an evolutionarily conserved gene essential for vertebrate development and ciliogenesis[27–29]. Recently, several groups independently identified mutations in Talpid3 as a cause of JBTS and lethal ciliopathies, such as hydrocephalus and short-rib polydactyly syndrome[30–35]. Our previous studies showed that Talpid3 localizes at the distal ends of both centrioles and regulates vesicle docking during ciliogenesis[36,37]. Although Talpid3 localizes to both centrioles, foregoing studies suggested that Talpid3 could play an important role in mother centriole maturation. In this study, in an effort to begin understanding the molecular basis of centriole asymmetry and maturation, we identify Talpid3 and C2CD3 as regulators of DCP removal. We find that the removal of DCPs is not required for centriole duplication, but it plays an essential role in centriole maturation. Remarkably, we show that Talpid3 regulates centriole maturation and ciliary vesicle docking through distinct regions. Furthermore, we show that removal of certain DCPs acts as an additional layer of DA assembly control, via regulation of OFD1 recruitment. Importantly, although the location and function of Talpid3, C2CD3, and OFD1 are coordinated and, in some cases, interdependent, we found that each protein exhibits distinct roles in regulating the actin network, CS organization, removal of DCPs, and assembly of appendages. Lastly, our data suggest potential mechanisms to explain how Talpid3 mutations found in JBTS contribute to disease phenotypes.

## Results

**Talpid3 regulates early and late centriole maturation events through distinct regions.** To gain further insight into the role of Talpid3 in ciliogenesis, we used CRISPR/Cas9-mediated gene editing to generate a Talpid3$^{-/-}$ retinal pigment epithelial (RPE1) cell line[37] and systematically observed the localization of DCPs, as well as markers of mother centriole/basal body maturation and ciliation, by immunofluorescence (IF). First, we found that Talpid3 KO cells recapitulated the defects caused by depletion or loss of Talpid3 using RNAi and genetic knock-outs[27–29,36], including the failure to dock ciliary vesicles and assemble primary cilia after serum withdrawal (Fig. 1a). Consistent with previous studies[34,36], we also observed elongated centrioles in Talpid3$^{-/-}$ cells using electron microscopy (EM; Fig. 1b). Moreover, we found that DA proteins (CEP83, CEP89, CEP164, and FBF1) were not observed at centrosomes in Talpid3 knock-out cells (Fig. 1a). Consistent with the role of DA in multiple processes, we found that IFT protein (IFT88 and IFT140) localization, TTBK2 recruitment, and CP110 removal were abrogated in Talpid3$^{-/-}$ cells. In each case, we showed that these results could not be explained by altered protein levels (Fig. 1c), leading us to conclude that recruitment per se was impacted. We note that CP110 and CEP164 recruitment persisted in prior Talpid3 knock-down experiments[36]. Since residual Talpid3 persists in siRNA-treated cells, our data suggest that recruitment of DA proteins is very sensitive to Talpid3 dosage and that recruitment of these proteins fails completely in the absence of Talpid3. On the other hand, we observed that assembly of SDA was intact in knock-out cells, similar to siRNA-depleted cells. Examination of Talpid3$^{-/-}$ cells by EM after serum withdrawal confirmed the absence of DA, ciliary vesicle docking, and ciliation, while confirming that normal SDA were assembled (Fig. 1b).

Remarkably, we also observed abnormal localization of DCPs (CEP120, Centrobin, and Neurl4) in Talpid3 KO cells: rather than exhibiting asymmetric enrichment on daughter centrioles, DCPs were found on both mothers and daughters (Fig. 1a). Asymmetric localization of DCPs is maintained by removal of DCPs from daughter centrioles during the G1–S transition, when centriole duplication is initiated[5,6]. However, DCP removal was completely blocked in Talpid3$^{-/-}$ cells (Supplementary Fig. 1a). Since centriole duplication is grossly normal in Talpid3$^{-/-}$ cells (Supplementary Fig. 1a), these data also suggest that removal of DCPs is not required for centriole duplication, but it might be involved in other processes. We showed that each of these phenotypes could be rescued by reintroducing full-length Talpid3 into Talpid3$^{-/-}$ cells, confirming that the observed defects were provoked specifically by the loss of Talpid3 (Fig. 2a). To identify regions in Talpid3 required for each stage of maturation and ciliogenesis, we performed rescue experiments in Talpid3$^{-/-}$ cells with a series of Talpid3 truncations. Strikingly, our data suggest that a region encompassing amino acids 466–700 is required for asymmetric localization of DCPs (Neurl4, Centrobin, and CEP120; Fig. 2a and Supplementary Fig. 1b), DA assembly (CEP83), and IFT complex recruitment (IFT88; Fig. 2b). We further confirmed the rescue of DA assembly by examining CEP164 rings using 3D-structured illumination microscopy (SIM, Supplementary Fig. 1c). In contrast, visualization of Rab8a and SmoM2, two early ciliary vesicle markers, or CP110 indicated that residues 701–1533 are essential for vesicle docking, CP110 removal, and ciliogenesis (GT335) (Fig. 2a, b). Importantly, these

data also (1) indicate that assembly of DA is sufficient for recruitment of the IFT machinery but not for initiation of vesicle docking or subsequent events during ciliogenesis and (2) pinpoint

a requirement for separable regions of Talpid3 in the maturation of mother centrioles/basal bodies versus the coordinated docking of vesicles and CP110 removal. In other words, a single protein,

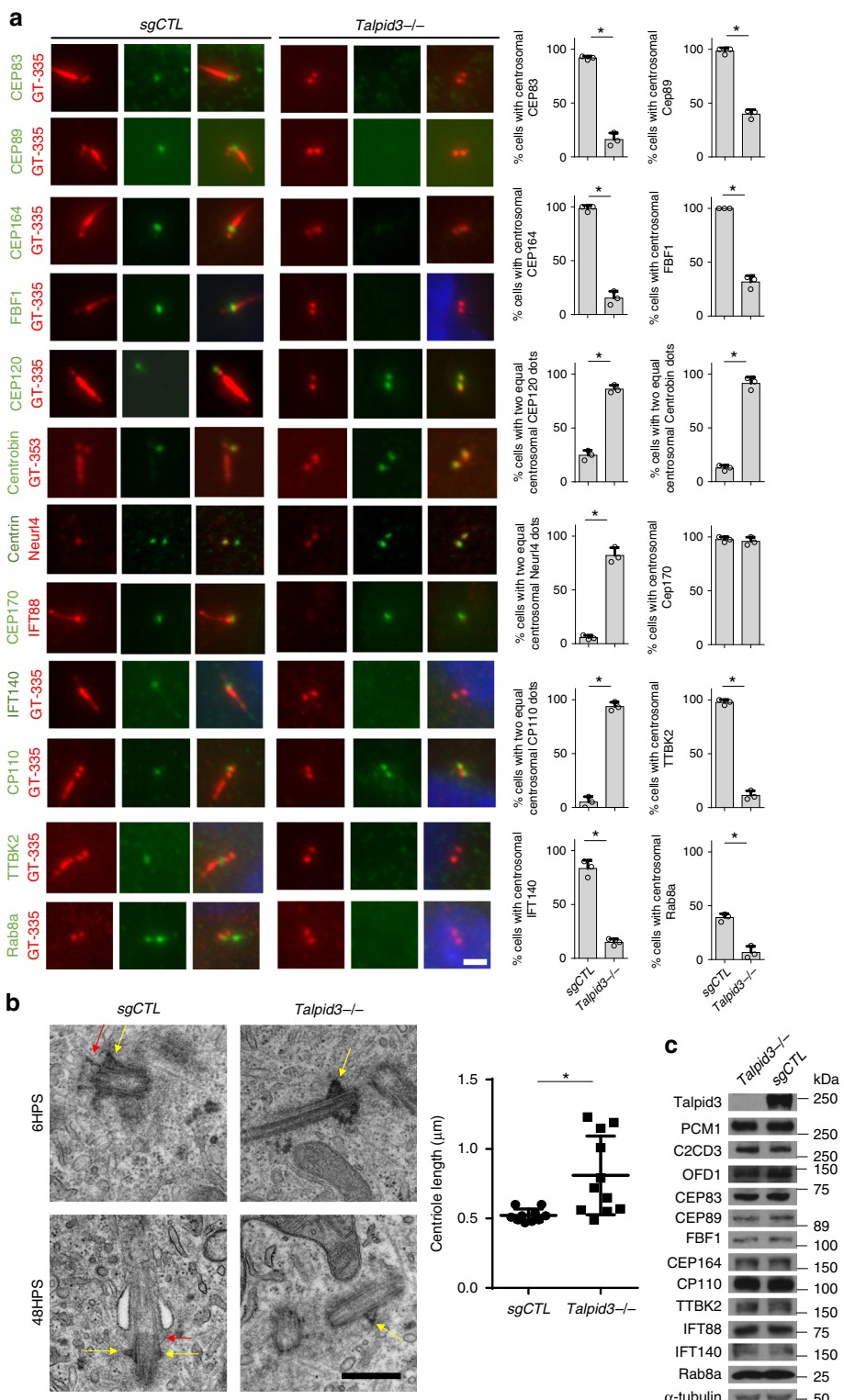

**Fig. 1** Talpid3 regulates centriole maturation and vesicle docking. **a** Centrosomal proteins and vesicular proteins were examined in control and *Talpid3*$^{-/-}$ cells by IF after 6 (Rab8a) or 48 h (all markers except for Rab8a) of serum-starvation using indicated antibodies. Scale bar = 2 μm. The protein level of these markers was examined by western blot (WB) in **c**. Cumulative data from three independent experiments are shown. For each group, a minimum of 100 cells/experiment was averaged. **b** Centrosome and primary cilium structure in control and *Talpid3*$^{-/-}$ cells was examined by transmission electron microscopy (TEM) after 6 and 48 h of serum-starvation (HPS). Red arrows indicate DA/transitional fibers. Yellow arrows indicate SDA. Data from one experiment are shown (*N* = 10 for *sgCTL* and *N* = 11 for *Talpid3*$^{-/-}$). All data are presented as mean ± SD. *$p$ < 0.05 (unpaired *t*-test). Scale bar = 0.5 μm

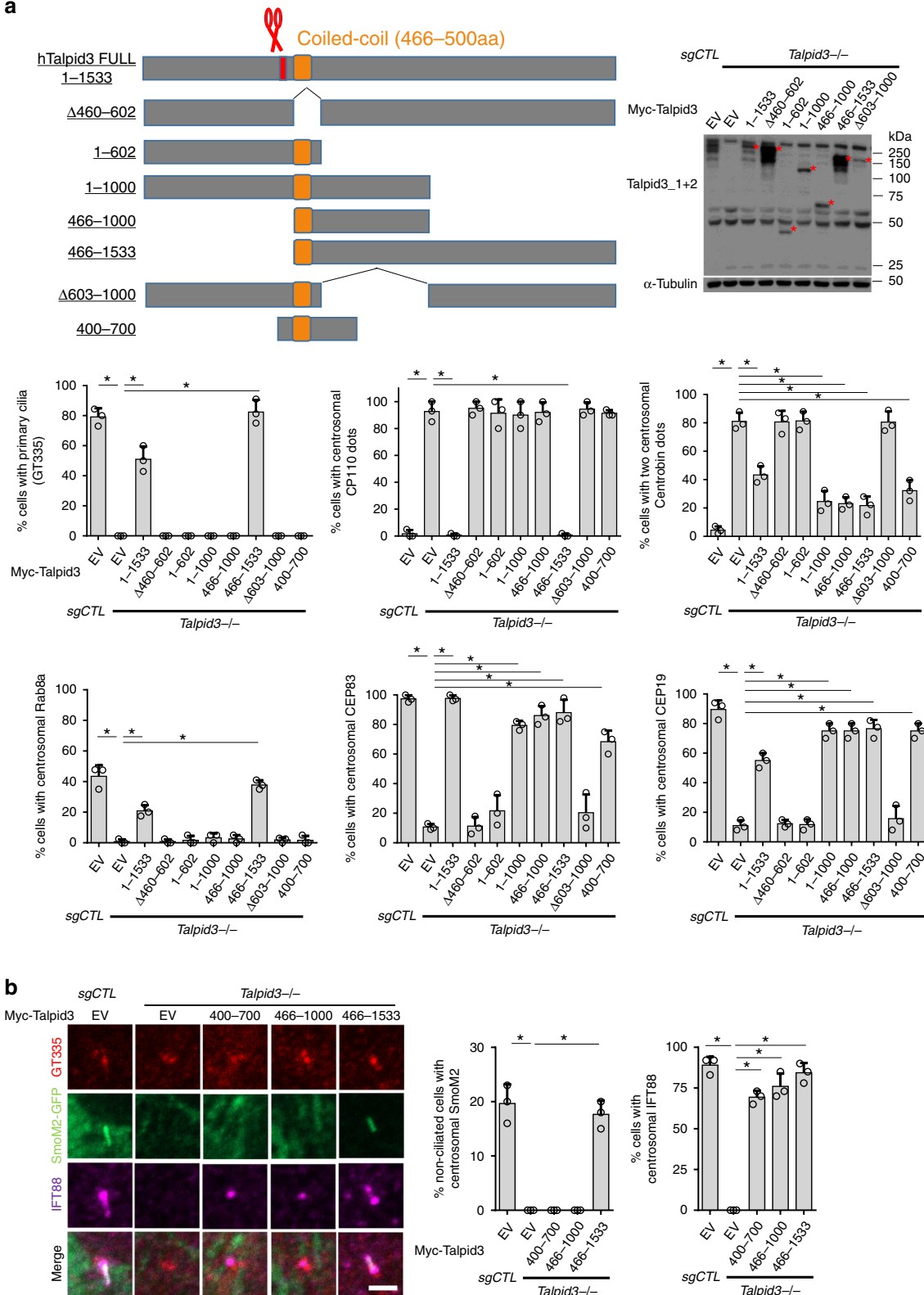

**Fig. 2** Talpid3 regulates centriole maturation and vesicle docking through distinct regions. **a** Centrosomal and ciliary defects of $Talpid3^{-/-}$ cells were rescued by infection with lentiviruses expressing different Myc-tagged Talpid3 constructs. Cells were serum-starved for 48 h and examined by IF with antibodies against indicated antibodies. Coiled-coil domain and CRISPR-targeted region of $Talpid3^{-/-}$ cells are shown in orange and red (scissors), respectively. Specific truncation proteins are indicated with an asterisk. The fragment spanning residues 400–700 is not recognized by Talpid3 antibodies and thus was detected by Myc tag, shown in Supplementary Figure 1b. **b** Stable $Talpid3^{-/-}$ cell lines expressing Talpid3 truncations were transduced with lentiviruses expressing SmoM2-GFP. Two days after transduction, cells were serum-starved for 6 h and stained with indicated antibodies. Cumulative data from three independent experiments are shown. For each group, a minimum of 100 cells/experiment was averaged. All data are presented as mean ± SD. *$p < 0.05$ (unpaired $t$-test). Scale bar = 2 μm

Talpid3, coordinates three activities essential for ciliogenesis: removal of DCP, assembly of DA/maturation of basal bodies, and recruitment of ciliary vesicles.

**Removal of specific DCPs is a prerequisite for DA assembly**. We were particularly intrigued by the centriole maturation defects observed in $Talpid3^{-/-}$ cells for two reasons. First, DA assembly is thought to be the earliest obligatory step for the initiation of ciliogenesis[38], and the major defects observed in $Talpid3^{-/-}$ cells, including the failure to dock vesicles, TTBK2 recruitment, CP110 removal, and IFT transport could be attributed to the DA assembly defect in Talpid3 null cells. Secondly, little is known about the regulatory mechanisms underpinning the asymmetric localization of DCPs. We speculated that during the maturation of daughter centrioles, removal of DCPs at the G1/S transition could be a prerequisite for the acquisition of appendages occurring later in the G2 phase. To test this hypothesis, DCPs were forced to symmetrically localize on both centrioles in wild-type RPE1 cells by expressing fusion proteins containing the PACT domain[39]. We found that PACT–CEP120 and PACT–Centrobin, but not PACT–Neurl4, were able to disrupt the localization of DA proteins, CEP83 and CEP164 (Fig. 3a and Supplementary Fig. 2a). These results suggest that the persistence of specific daughter centriole proteins is sufficient to suppress DA assembly.

Conversely, to further examine whether the failure to remove CEP120 and Centrobin inhibits DA assembly, we knocked down Centrobin in $Talpid3^{-/-}$ cells using siRNAs. Strikingly, we found that depletion of Centrobin was able to substantially rescue the assembly of DA (Fig. 3b and Supplementary Figs. 2a, b). The rescue of DA assembly was further confirmed by measuring the diameter of CEP164 rings using SIM (Supplementary Fig. 2c). We note that rescue of DA assembly was unable to restore ciliogenesis in Talpid3 KO cells, consistent with an additional role(s) of Talpid3 in ciliogenesis beyond DCP removal and DA assembly. We also silenced CEP120, but knocking down this protein blocked centriole duplication and resulted in cells with one centriole or no centrioles (Supplementary Fig. 2b and Fig. 3d), as expected[5], preventing us from confirming its role in inhibiting DA assembly. We next asked whether there was a reciprocal relationship between DA assembly and DCP appearance. However, disruption of DA assembly through depletion of CEP83 had no effect on the asymmetric localization of DCPs in wild type RPE1 cells (Fig. 3c). We found that DCPs were recruited in a sequential manner, with CEP120 localization required for recruitment of Centrobin, and recruitment of Neurl4 dependent upon both CEP120 and Centrobin, suggesting that removal of DCPs could also occur sequentially. Moreover, we found that Centrobin is required for maintenance of CEP120 asymmetry, since depletion of Centrobin led to symmetric localization of CEP120 on both centrioles (Fig. 3d). These data reveal a robust network underlying the daughter-to-mother centriole transition, wherein the removal of DCPs, Centrobin, and perhaps CEP120, is a prerequisite for DA assembly but not vice versa.

**Talpid3 and C2CD3 coordinately regulate mother centriole maturation**. Next, we investigated how Talpid3 regulates the asymmetric localization of DCPs, and we considered several possibilities. First, since DCPs were recruited in a sequential manner, Talpid3 could regulate CEP120 localization through direct interactions with CEP120, as suggested previously[40]. We tested the interaction between different Talpid3 truncations and CEP120 and found that CEP120 interacted most robustly with a fragment spanning residues 466–1000, although this protein

interacted with multiple surfaces of Talpid3 (Supplementary Fig. 3). We found that the localization of Talpid3 to centrioles was important for this interaction, since a mutant protein from a JBTS patient (C.1697A>T) that failed to localize to centrosomes (see below) exhibited significantly impaired interactions with CEP120. Importantly, since the 1–602 fragment of Talpid3 was able to interact with CEP120 without rescuing the centriole maturation defects observed in $Talpid3^{-/-}$ cells, we conclude that Talpid3–CEP120 interactions are not sufficient to regulate asymmetric localization of DCPs. Secondly, since CS disorganization is observed in $Talpid3^{-/-}$ cells, and Talpid3 was shown to be required for vesicle docking[36], it was possible that Talpid3 could regulate centriole maturation through CS. To test this possibility, we utilized the $PCM1^{-/-}$ RPE1 cell line[37], which lacks CS. We found that the asymmetric localization of DCPs and DA assembly were normal in $PCM1$ KO cells, demonstrating that the formation of CS is not required for these two events (Fig. 4a).

We speculated that Talpid3 could also regulate the asymmetric localization of DCPs through other distal centriolar proteins. C2CD3 and OFD1 have been shown to regulate DA assembly, although their mechanisms remain obscure, and thus we investigated these candidates as regulators of asymmetric localization of daughter centriole proteins. We generated $C2CD3^{-/-}$ and $OFD1^{-/-}$ cell lines using CRISPR/Cas9 and found that, consistent with previous reports[24–26], both proteins were required for DA assembly, and these defects could be rescued with full-length C2CD3 and OFD1, respectively (Fig. 4a, Supplementary Fig. 4a). Interestingly, $C2CD3^{-/-}$, but not $OFD1^{-/-}$ cells, displayed symmetric localization of CEP120 and Centrobin (Fig. 4a, b), but this was not due to increased protein levels (Supplementary Fig. 4b). Strikingly, depletion of Centrobin could partially rescue the localization of DA proteins in $C2CD3^{-/-}$, but not in $OFD1^{-/-}$ cells (Fig. 4b). These data not only confirmed our previous conclusion that removal of certain DCPs is a prerequisite for DA assembly, but they also suggested that C2CD3 and Talpid3 control asymmetric localization of DCPs, whereas OFD1 may be required primarily for DA assembly. Since maturation occurs during the G2/M phase, it was possible that the defects in maturation could result from cell cycle arrest prior to this stage. However, we confirmed that the centriole maturation defects found in $Talpid3^{-/-}$, $C2CD3^{-/-}$, and $OFD1^{-/-}$ cells did not arise from aberrant cell cycle progression or the altered abundance of daughter centriole proteins (Supplementary Fig. 4b).

We next explored the functional interactions between Talpid3, C2CD3, and OFD1 in greater detail. First, to determine whether recruitment of these proteins occurred sequentially, we examined their localization in all three KO cell lines. We found that Talpid3 and C2CD3 were coordinately recruited to mother and daughter centrioles, and both were also required to recruit OFD1 (Fig. 4c). In contrast, OFD1 was not required to recruit either Talpid3 or C2CD3. Further, we found that Talpid3 could interact most robustly with C2CD3 through residues 466–1533, whereas amino-terminal portions of Talpid3 were unable to do so (Fig. 4d). Taken together with previous data suggesting a functional interaction between C2CD3 and OFD1, we conclude that Talpid3, C2CD3, and OFD1 form a complex at the distal ends of centrioles.

To extend our understanding of the critical role of C2CD3 and OFD1 recruitment by Talpid3, we examined the ability of Talpid3–C2CD3 and Talpid3–OFD1 fusion proteins to functionally reconstitute each step of the maturation process in $Talpid3^{-/-}$ cells. We fused the Talpid3 amino-terminal 1–602 aa fragment (Talpid3Nter), which is required for centrosome localization but is not sufficient for centriole maturation[36] (Fig. 2a), with C2CD3 or OFD1, thereby tethering each protein to centrioles in $Talpid3$ KO cells. We observed that tethering of

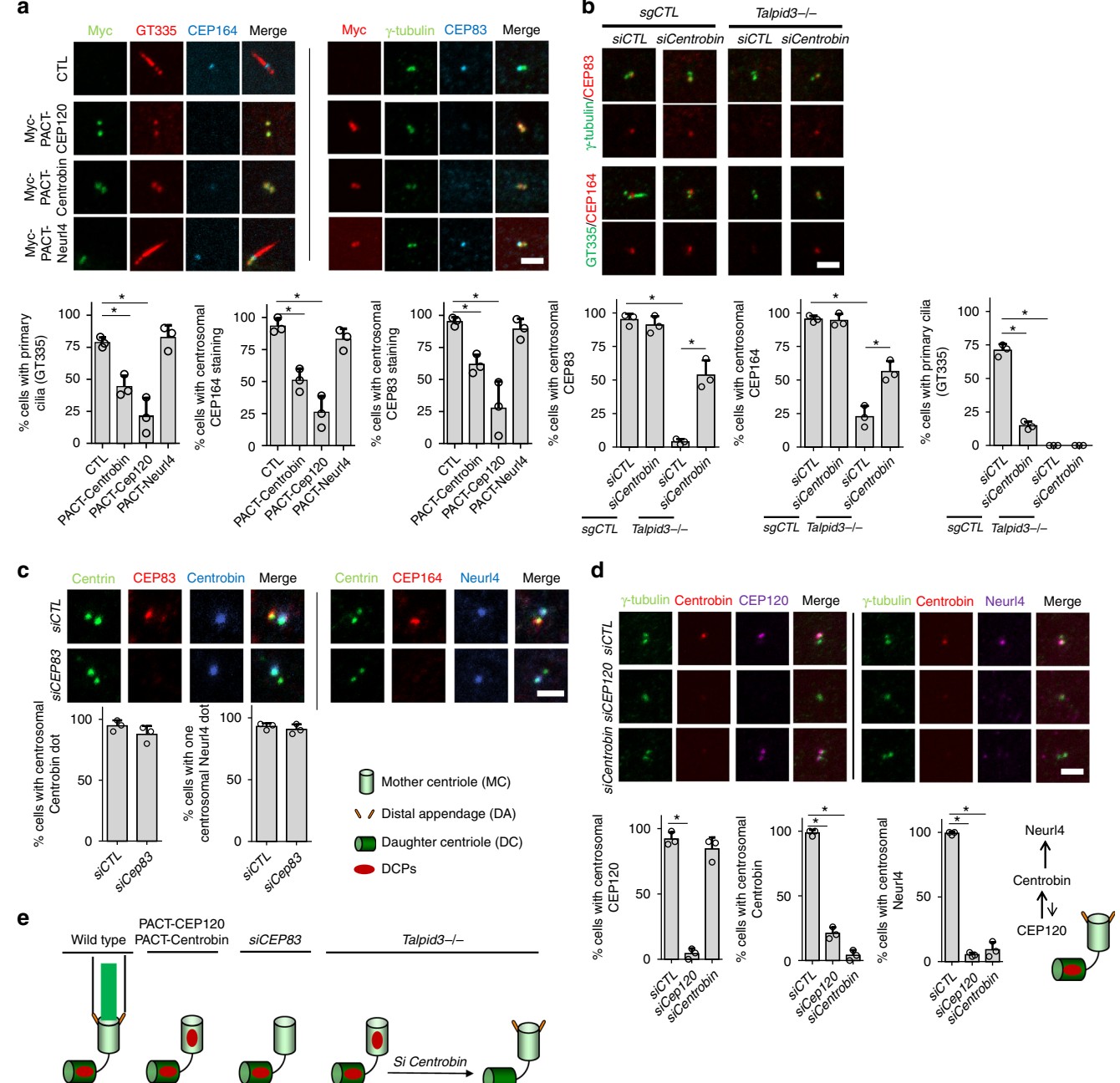

**Fig. 3** Removal of specific DCPs is a prerequisite for DA assembly. **a** WT RPE1 cells were transduced with lentiviruses expressing Myc-PACT–CEP120, Myc-PACT–Centrobin, or Myc-PACT-Neurl4, as indicated. Two days after transduction, cells were serum-starved for 48 h and examined by IF with antibodies against DA markers, CEP83 and CEP164. **b** Control and *Talpid3−/−* cells were transfected with siRNAs against CEP120 (Supplementary Figures 2a, b) and Centrobin. Two days after transfection, cells were serum-starved for 24 h and examined by IF and WB using indicated antibodies. **c** WT RPE1 cells were transfected with siRNAs against CEP83. Two days after transfection, cells were serum-starved for 24 h and were visualized with indicated antibodies to check DA and DCPs. **d** WT RPE1 cells were transfected with siRNAs against CEP120 and Centrobin. Two days after transfection, cells were serum-starved for 24 h and then visualized with indicated antibodies to check the localization of DCPs, which are recruited as schematized (right). **e** Schematic of the relationship between removal of DCPs and DA assembly. Cumulative data from three independent experiments are shown. For each group, a minimum of 100 cells/experiment was averaged. All data are presented as mean ± SD. *$p < 0.05$ (unpaired *t*-test). Scale bars = 2 μm

C2CD3 could substantially rescue defective recruitment of DA proteins in *Talpid3* KO cells (Fig. 4e). Tethering of OFD1 could also partially rescue the recruitment of DA proteins, although its impact was considerably less robust than that of C2CD3. These data suggest that centriolar recruitment of C2CD3 by Talpid3 plays an important role in the regulation of asymmetric localization of DCPs, proper OFD1 localization, and subsequent DA assembly. Indeed, our data suggest that a key role for Talpid3

is the targeting and recruitment of other proteins, C2CD3 and OFD1, to the distal end and that such recruitment can largely bypass the loss of Talpid3.

**Asymmetric localization of DC-enriched proteins is required for proper localization of OFD1.** Given that abnormal, symmetric localization of DCPs is accompanied by the absence of

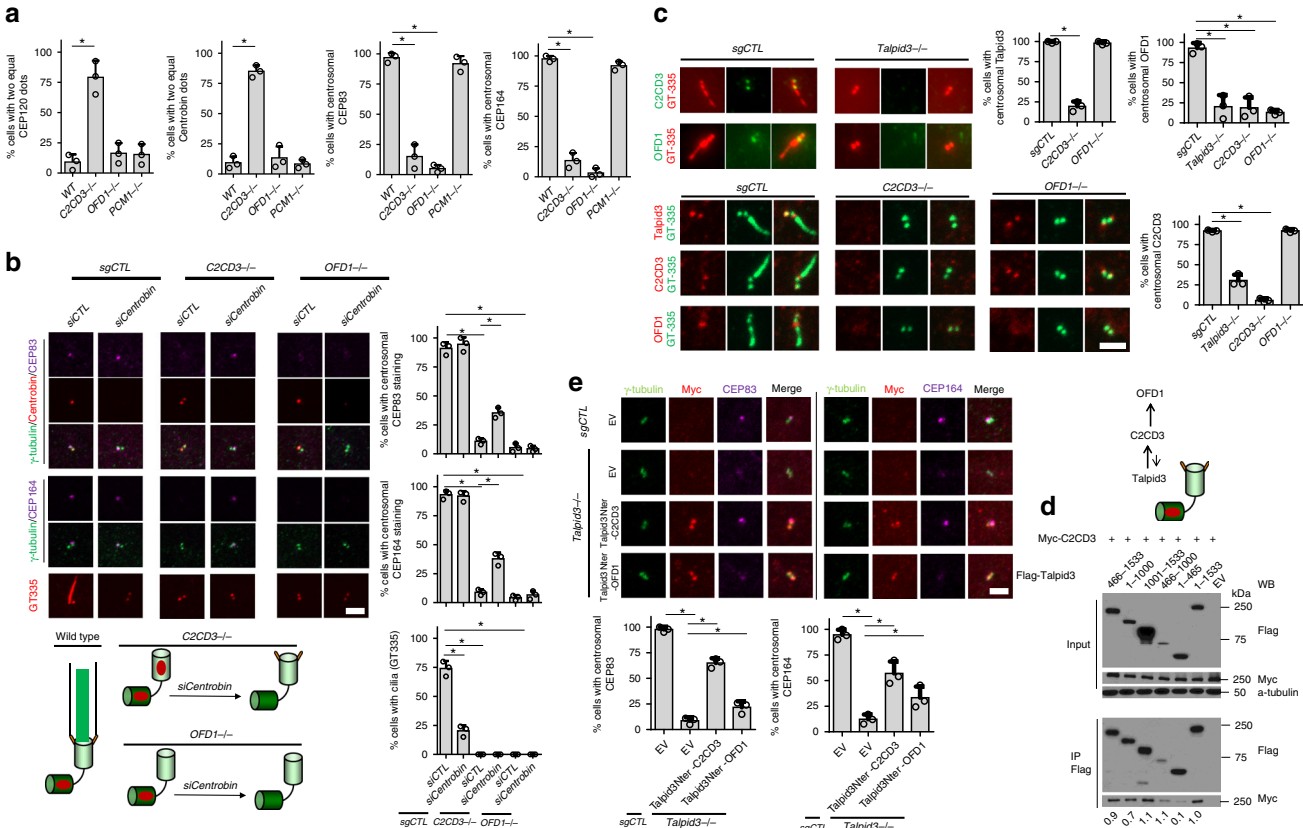

**Fig. 4** Talpid3 and C2CD3 coordinately regulate mother centriole maturation. **a** Localization of DCPs and DA proteins were checked in control, $C2CD3^{-/-}$, $OFD1^{-/-}$, and $PCM1^{-/-}$ cells. Cells were serum-starved for 24 h and then visualized with indicated antibodies. **b** Control, $C2CD3^{-/-}$, and $OFD1^{-/-}$ cells were transfected with siRNAs against CEP83. Two days after transfection, cells were serum-starved for 24 h and were visualized with indicated antibodies to examine DA and DCPs. **c** Localization of Talpid3, C2CD3, and OFD1 was checked in control, $C2CD3^{-/-}$, $OFD1^{-/-}$, and $Talpid3^{-/-}$ cells. Cells were serum-starved for 24 h and then visualized with indicated antibodies. **d** 293T cells stably expressing N-terminally Myc-tagged C2CD3 and N-terminally Flag-tagged Talpid3 were immunoprecipitated with anti-Flag antibody. Eluates were analyzed by immunoblotting with indicated antibodies. Numbers at the bottom of each lane represent the quantification of band intensities of immunoprecipitated Flag-CEP120, which was first normalized to the lane containing the full-length Talpid3 protein (1–1533) and then normalized to the intensities of each corresponding, co-immunoprecipitated Myc-Talpid3 truncation. **e** DA assembly in $Talpid3^{-/-}$ cells was substantially or partially rescued by infection with lentiviruses expressing an amino-terminal fragment of Talpid3 (residues 1–602) fused to C2CD3 or OFD1. Cells were examined by IF after 48 h of serum starvation using the indicated antibodies. Cumulative data from three independent experiments are shown. For each group, a minimum of 100 cells/experiment was averaged. All data are presented as mean ± SD. *$p < 0.05$ (unpaired $t$-test). Scale bars = 2 μm

OFD1 from centrioles in Talpid3 and C2CD3 KO cells (Figs. 1a and 4a, c), we hypothesized that the failure to remove DCPs during centriole maturation could prevent centrosomal localization of OFD1 and thus assembly of DA. To test our hypothesis, DCPs were again forced to symmetrically localize on both centrioles by expressing PACT domain fusions in wild-type RPE1 cells. We found that PACT–CEP120 and PACT–Centrobin, but not PACT–Neurl4, were able to disrupt the localization of OFD1 (Fig. 5a). To further examine whether failure to remove Centrobin plays an inhibitory role in OFD1 recruitment, we knocked down Centrobin in $Talpid3^{-/-}$ and $C2CD3^{-/-}$ cells with siRNAs and found that the localization of OFD1 was largely rescued (Fig. 5b). The rescue of centrosomal OFD1 was further confirmed by measuring the diameter of OFD1 rings using SIM (Supplementary Fig. 2c). Since tethering of C2CD3 to centrioles in $Talpid3^{-/-}$ cells using an amino-terminal fusion could rescue the recruitment of DA proteins (Fig. 4e), we anticipated that the recruitment of OFD1 would also be rescued in these cells, and this was indeed the case (Fig. 5c). In total, these data demonstrate that the failure to remove DCPs blocks centrosomal OFD1 recruitment and thus abrogates DA assembly.

In addition to their centriolar localization, C2CD3 and OFD1 also partition to CS, and this could potentially contribute to centriole maturation. To address whether the CS pool of C2CD3 and OFD1 plays a role in centriole maturation, we first examined the CS localization of C2CD3 and OFD1 in $PCM1^{-/-}$ cells using Myc-tagged C2CD3 and an anti-OFD1 antibody (OFD1–2) that are able to detect CS pools of C2CD3 and OFD1, respectively. Compared with control cells, the CS pools of C2CD3 and OFD1 were depleted in $PCM1^{-/-}$ cells (Fig. 4a), and only centrosomal C2CD3 and OFD1 remained (Supplementary Fig. 5a). Since centriole maturation is normal in $PCM1^{-/-}$ cells, these data suggest that the CS pools of C2CD3 and OFD1 are not required for centriole maturation. Moreover, we found that PACT–CEP120, PACT–Centrobin, and PACT–Neurl4 do not affect the CS pool of OFD1 (Supplementary Fig. 5b). Together, these data demonstrate that centriole-bound Talpid3 and C2CD3 regulate the removal of DCPs, which, in turn, controls the centrosomal recruitment of OFD1 and DA assembly (Fig. 5d).

**Spatial coordination of Talpid3/C2CD3/OFD1 complex with DC protein asymmetry and DA assembly.** To better understand the spatial basis for regulation of asymmetric localization of DC

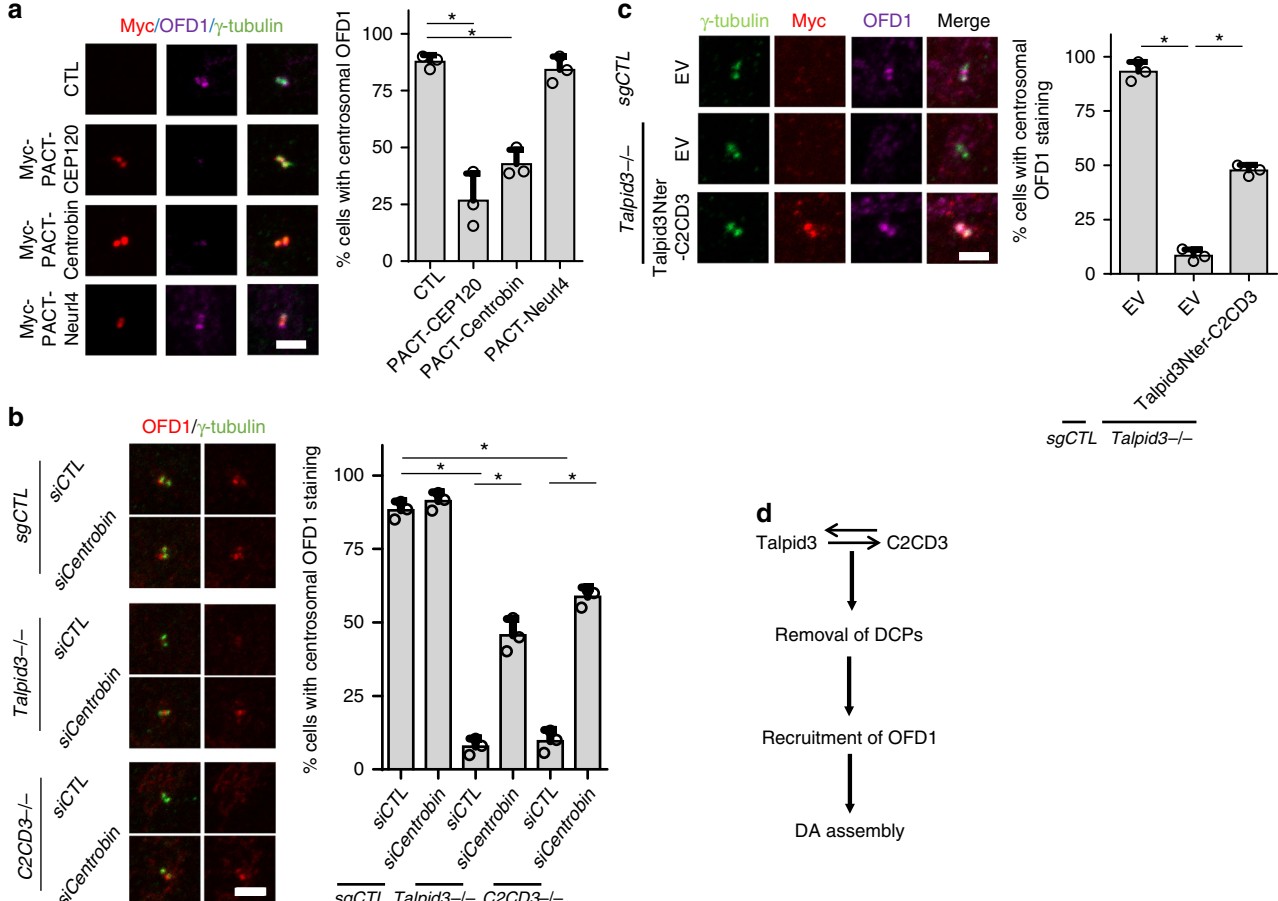

**Fig. 5** Asymmetric localization of DC-enriched proteins is required for proper localization of OFD1. **a** WT RPE1 cells were transduced with lentiviruses expressing Myc-PACT–CEP120, Myc-PACT–Centrobin, or Myc-PACT-Neurl4, as indicated. Two days after transduction, cells were serum-starved for 48 h and examined by IF with indicated antibodies. **b** Control, $C2CD3^{-/-}$, and $Talpid3^{-/-}$ cells were transfected with siRNAs against Centrobin. Two days after transfection, cells were serum-starved for 24 h and were visualized with indicated antibodies. **c** OFD1 recruitment defect in Talpid3$^{-/-}$ cells was rescued by infection with lentiviruses expressing the Talpid3–C2CD3 fusion protein described in Fig. 3e. Cells were examined by IF after 48 h of serum starvation using indicated antibodies. **d** Model indicating a centriole maturation regulatory mechanism. Cumulative data from three independent experiments are shown. For each group, a minimum of 100 cells/experiment was averaged. All data are presented as mean ± SD. *$p < 0.05$ (unpaired $t$-test). Scale bars = 2 µm

proteins instigated by Talpid3 and C2CD3, we investigated the compartmentalization of distal end proteins and DCPs (Talpid3, C2CD3, CEP120, and Centrobin) using SIM (Fig. 6a). We observed that C2CD3 localized to a small dot at the extreme distal end of both centrioles, enveloped by a Talpid3 ring. Centrobin formed a ring in a middle segment of the daughter centriole, adjacent to the Talpid3 ring and distal to the proximal marker, GT335. Consistent with a previous report[1], CEP120 was distributed along the length of the DC centriole barrel, and it co-localized with Talpid3 at the distal end, in line with data indicating that these proteins interact (Fig. 6a and Supplementary Fig. 3). Our data demonstrate that a complex consisting of Talpid3 and C2CD3 at the distal end of centrioles partially co-localizes with CEP120 and is adjacent to the Centrobin ring, prompting speculation that the asymmetric localization of DC-enriched proteins could be controlled at the distal end of centrioles. OFD1 also displayed different localization patterns on the mother (MC) and daughter (DC) centrioles (Fig. 6a). On the DC, OFD1 formed a ring at the distal end that co-localized with the Talpid3 ring. On the MC, OFD1 formed a much larger ring at the distal end that co-localized with, and had the same diameter as, the CEP164 ring. These data suggest that OFD1 may be peripherally associated with DA structure and could play a more direct

role in DA assembly. We also examined the localization of CEP120 and Centrobin using SIM to determine how these DCPs partitioned in $Talpid3^{-/-}$, $C2CD3^{-/-}$, and $PCM1^{-/-}$ cells. Using CP110 and GT335 as centriolar distal and proximal markers, respectively, we observed that CEP120 and Centrobin staining appeared identical along the barrels of both centrioles, from the proximal to distal ends, in $Talpid3^{-/-}$ and $C2CD3^{-/-}$ cells (Fig. 6b). In contrast, Cep120 and Centrobin partitioned preferentially to DC in control and $PCM1^{-/-}$ cells. These data unequivocally demonstrate that removal of DCPs was completely blocked in $Talpid3^{-/-}$ and $C2CD3^{-/-}$ cells, resulting in a symmetrical distribution of DCPs on both centrioles. Interestingly, we found that PACT-domain fusions with DCPs localized along the barrels of both centrioles, mimicking the localization of CEP120 and Centrobin in $Talpid3^{-/-}$ and $C2CD3^{-/-}$ cells (Fig. 6b and Supplementary Fig. 5c). Specifically, the diameters of PACT–CEP120 (384 ± 34 nm) and PACT–Centrobin (395 ± 26 nm) rings are comparable to those of endogenous CEP120 (378 ± 29 nm) and Centrobin (356 ± 33 nm) proteins, which are consistent with previous reports showing that CEP120 and Centrobin localize close to the outer centriole wall[5,6,41]. These data suggest that the "default" state for localization of DCPs could be on or near the barrels of both centrioles, but Talpid3 and C2CD3 act to

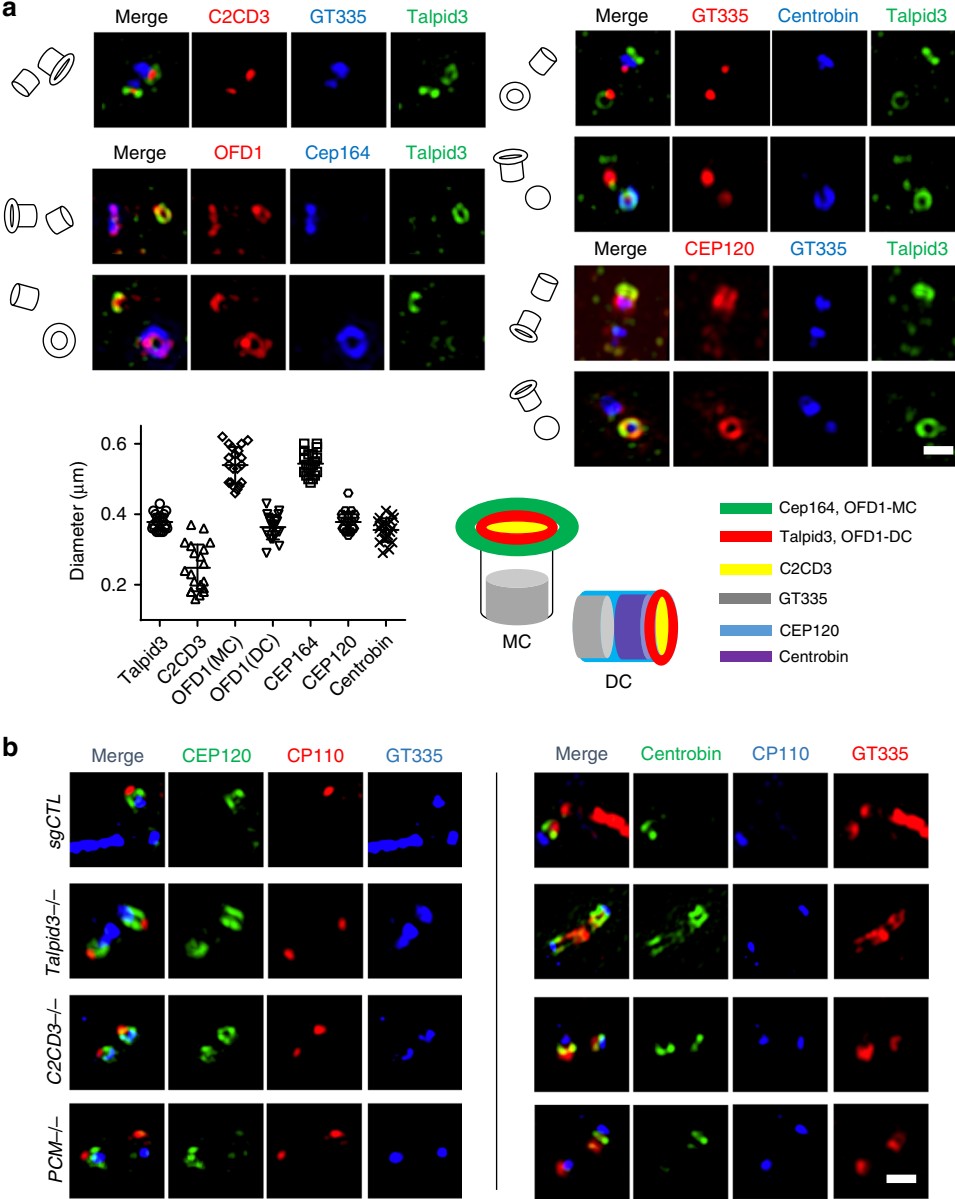

**Fig. 6** Topography of distal and daughter centriolar proteins. **a** Growing RPE1 cells were stained with the indicated combinations of antibodies and visualized using structured illumination microscopy (SIM). Bottom left: diameter of the Talpid3/C2CD3/OFD1 complex, DCPs, and DA protein, CEP164 (*N* = 20). Cumulative data from two independent experiments are shown. Bottom right: schematic representation of the centrosome illustrates the localization of Talpid3/C2CD3/OFD1 complex, DCPs, and DA protein, CEP164. **b** Localization of CEP120 and Centrobin was examined in control, *Talpid3*$^{-/-}$, *C2CD3*$^{-/-}$, and *PCM1*$^{-/-}$ cells using SIM. Cells were serum-starved for 24 h and then visualized with indicated antibodies. Scale bars = 0.5 μm

promote removal of DCPs from the mother centriole. Lastly, since ablation of Talpid3 leads to aberrantly long centrioles, and loss of C2CD3 leads to abnormally short centrioles (Figs. 1b and 6b[24]), our data suggest that the observed centriole maturation defects are not solely due to excessively long centrioles.

**Unique functions for Talpid3, C2CD3, and OFD1.** Our findings provided definitive evidence to support a role for a distal centriolar network in promoting loss of DCPs as well as assembly of DA, major hallmarks of organelle asymmetry. To further explore the function of Talpid3, C2CD3, and OFD1 in the acquisition of centriolar asymmetry and the assembly of distal ends, we examined SDA assembly in all three knock-outs. We observed that C2CD3, but not Talpid3 or OFD1, was required for proper localization of SDA proteins (Figs. 7a and 1b), since the

percentage of *C2CD3*$^{-/-}$ cells with centrosomal ODF2, CEP128, and Centriolin decreased by ~60% as compared to controls (Fig. 7a), consistent with EM studies in *C2cd3* and *Ofd1* mutant mouse embryonic fibroblasts (MEFs)[24,26]. In contrast, removal of CEP128 and disruption of SDA did not affect the localization of Talpid3, C2CD3, and OFD1. These data also suggest that assembly of SDA is independent of DCP removal and DA assembly.

Next, we examined the centrosomal localization of the recently identified CEP350/FOP/CEP19 module[42–44] in all three KO cell lines. On mother centrioles, the CEP350/FOP/CEP19 complex partitions to a region near the SDA (CEP350/FOP) or between sub-distal and distal (CEP19) appendages, and it is essential for IFT trafficking and ciliogenesis. Further, whereas CEP350 and FOP can be recruited to either centriole, CEP19 is enriched at the

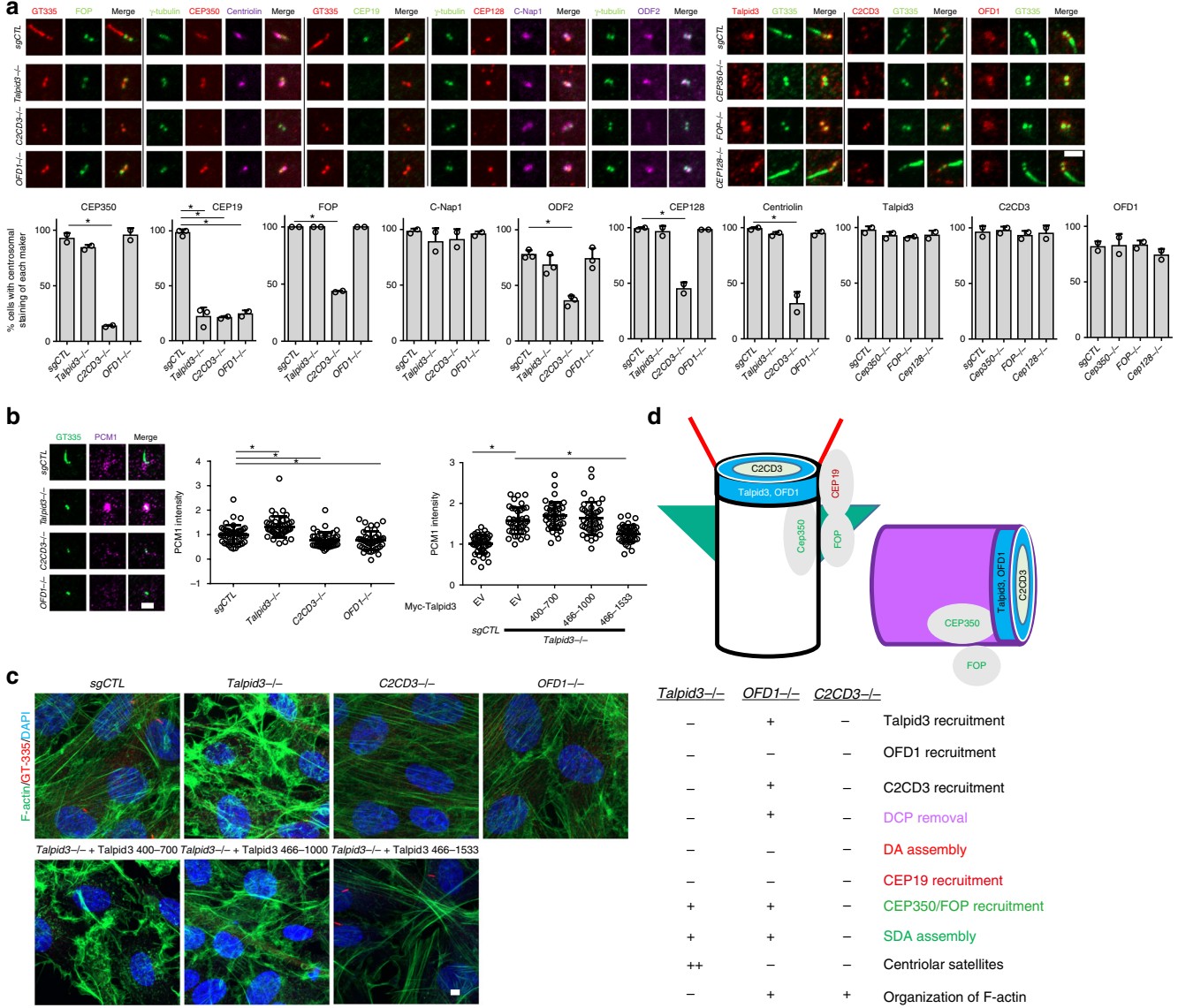

**Fig. 7** Unique functions for Talpid3, C2CD3, and OFD1 in assembling distal-end structures. **a** Localization of CEP350/FOP/CEP19 module, SDA proteins, and **b**, **c** the organization of cytoplasmic actin and centriolar satellites (CS) was examined in control, C2CD3⁻/⁻, OFD1⁻/⁻, and Talpid3⁻/⁻ cells. Localization of Talpid3, C2CD3, and OFD1 was examined in control, CEP350⁻/⁻, FOP⁻/⁻, and CEP128⁻/⁻ cells. Cells were serum-starved for 24 h and then visualized with indicated antibodies. Cumulative data from two or three independent experiments are shown in **a**, as indicated by data dots. For each group, a minimum of 100 cells/experiment was averaged. Data from one experiment are shown in **b**, with 40 cells per group in the left panel and 45 cells per group in the right panel. Experiments were repeated independently two times with similar results. **d** Schematic of the unique functions for Talpid3, C2CD3, and OFD1 in assembling distal-end structures, CS and actin network (−, defective; +, normal; ++, enhanced). All data are presented as mean ± SD. *p < 0.05 (unpaired t-test). Scale bars = 2 μm

mother centriole/basal body upon serum starvation[45] (Fig. 7a). Strikingly, we observed that CEP19 localization was defective in all three KO cell lines, whereas CEP350 was absent only in C2CD3⁻/⁻ cells. Similarly, FOP staining was lost from ~60% of C2CD3⁻/⁻ cells, but it appeared normal in Talpid3⁻/⁻ and OFD1⁻/⁻ cells (Fig. 7a). In an effort to determine whether restoration of CEP19 could be tied to other mother centriole-specific maturation-associated events, we attempted to rescue its localization in Talpid3⁻/⁻ cells. Interestingly, residues 466–700 of Talpid3 were sufficient to restore proper localization of CEP19 in Talpid3 KO cells (Figs. 2a and 7a), indicating that recruitment of CEP19 may be tightly linked to critical steps in maturation, namely, removal of DCP and acquisition of DAs. The rescue of CEP19 localization was further confirmed by measuring the diameter of CEP19 rings using SIM

(Supplementary Fig. 1c). Conversely, ablation of CEP350 or FOP does not affect the localization of Talpid3, C2CD3, and OFD1. These data suggest that the Talpid3–OFD1–C2CD3 network, together with FOP and CEP350, play a prominent role in recruitment of CEP19. C2CD3 can be placed higher in a regulatory hierarchy by controlling recruitment of OFD1, FOP, and CEP350, and Talpid3 could regulate CEP19 localization through OFD1 (see Discussion).

Previous studies from our lab and others[36,46] suggested a functional antagonism between Talpid3 and OFD1 in the organization of CS. To confirm previous discoveries and compare the functions of Talpid3, C2CD3, and OFD1 in CS organization, we investigated the PCM1 staining pattern in Talpid3⁻/⁻, C2CD3⁻/⁻, and OFD1⁻/⁻ cells. Consistent with previous studies, we observed accumulation and increased intensity of CS near centrioles in

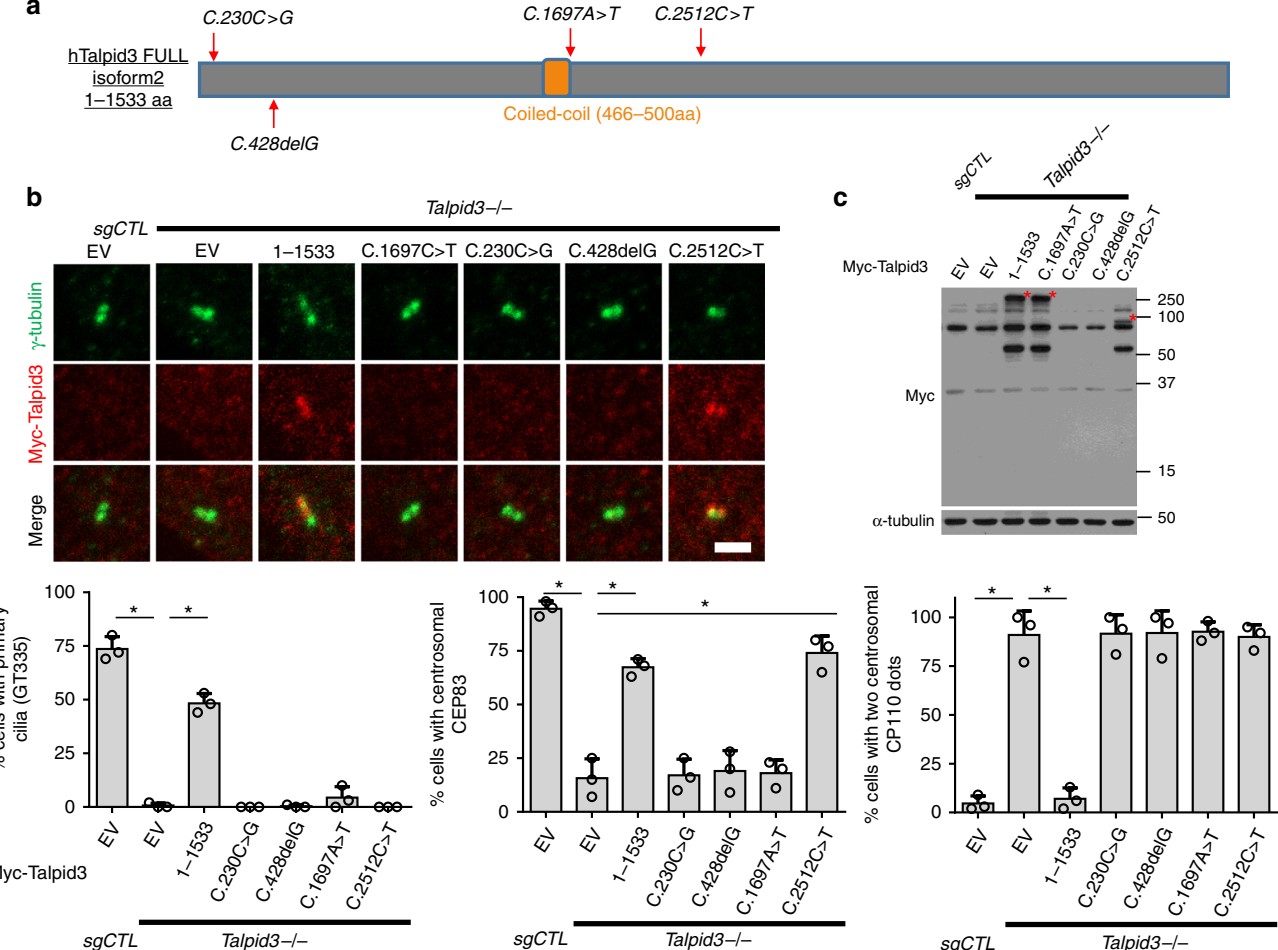

**Fig. 8** Talpid3 mutants associated with JBTS and other lethal ciliopathies display centriole maturation defects. **a** Summary of Talpid3 mutations under investigation and their locations (red arrows) in Talpid3 protein. Centrosomal and ciliary defects in *Talpid3* KO cells were rescued by infection with lentiviruses expressing Myc-tagged Talpid3 constructs. Cells were examined by IF (**b**) and WB (**c**) after 48 h of serum starvation using indicated antibodies. Specific truncation proteins are indicated with an asterisk. Cumulative data from three independent experiments are shown. For each group, a minimum of 100 cells/experiment was averaged. All data are presented as mean ± SD. *$p < 0.05$ (unpaired *t*-test). Scale bar = 2 μm

*Talpid3*$^{-/-}$ cells compared to controls (Fig. 7b), and in rescue experiments, residues 1001–1533 of Talpid3 were required to promote proper CS organization in null cells. Moreover, in *C2CD3*$^{-/-}$ and *OFD1*$^{-/-}$ cells, we observed a significant decrease in CS staining near centrioles (Fig. 7b). These data demonstrate that both C2CD3 and OFD1 are required for maintenance of CS organization, and C2CD3 may exert its role by regulating the localization of OFD1. These data also suggest that Talpid3 inhibits CS accumulation through other unknown regulators and that the absence of SDA or DA in these knock-out cells is not a result of aberrant organization of CS around centrosomes.

The distal ends of centrioles also play a role in assembling the actin network. This may be due in part to interactions between CP110, ciliary adhesion complexes, and components of the actin cytoskeleton[47,48]. In addition, loss of Talpid3 provokes remodeling of actin filaments in mouse and chicken mutants[28,49]. Indeed, we observed disorganization of the actin network, marked by reduced stress fibers and punctate staining in Talpid3 KO cells (Fig. 7c). In rescue experiments, the required regulatory region included residues 1001–1533 of Talpid3 (Fig. 7c). Interestingly, C2CD3 and OFD1 ablation did not disrupt actin organization (Fig. 7c). These studies suggest that Talpid3, a CP110-interacting protein, may be uniquely required among this group of distal proteins to organize actin networks. Moreover, our results suggest

that the absence of SDA or DA is not sufficient to promote remodeling of the actin cytoskeleton, and conversely, the ability to maintain the actin and CS networks does not guarantee the normal assembly of appendages.

Altogether, these data demonstrate that Talpid3, C2CD3, and OFD1 commonly regulate the assembly of DA, but they play distinct roles in DCP removal as well as in the assembly of SDA, CS, the actin cytoskeleton, and the CEP350/FOP/CEP19 module (Fig. 7d).

**Talpid3 mutations associated with ciliopathies affect centriole maturation and ciliogenesis.** Mutations in human *KIAA0586/Talpid3* have been linked to JBTS and other lethal ciliopathies[30–35]. It is not well understood how these mutations affect the centriolar structure and promote disease phenotypes. In an effort to determine which functions of Talpid3 are most critical, we expressed patient-derived *Talpid3* alleles and asked whether they are compromised for one or more of the functions that we have analyzed in our studies. We generated human Talpid3 constructs harboring a variety of mutations described previously and compared the ability of wild-type Talpid3 and four patient-derived mutant proteins to localize to the centriole and to rescue centriolar defects observed in

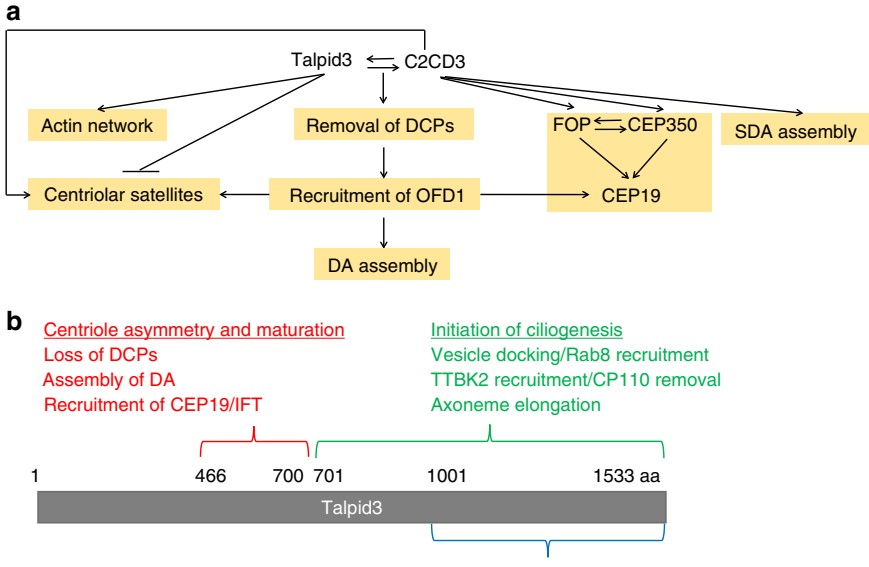

**Fig. 9** The Talpid3–C2CD3–OFD1 complex as a multi-functional hub that promotes centriole maturation and asymmetry. **a** Model summarizing how Talpid3, C2CD3, and OFD1 promote centriole maturation and assembly of distal structures. **b** Schematic of the functional and interacting regions of Talpid3. See text for details

Talpid3$^{−/−}$ RPE1 cells (Fig. 8). The substitution mutation, C.230C>G, and deletion mutation, C.428delG, are predicted to result in truncated proteins of 77 and 147 amino acids, respectively. The predicted truncations were not detected by immunoblotting, were unable to localize to centriole by IF, and were therefore unable to rescue centriole maturation or ciliogenesis (Fig. 8). The substitution mutation, C.1697A>T, causes a single amino acid change (p.D566V) near the conserved coiled-coil domain, which is required for centrosomal localization of Talpid3. Surprisingly, this mutant protein, though expressed at the same level as endogenous Talpid3, was unable to localize to centrosomes, and it was unable to rescue centriole maturation and ciliogenesis. Therefore, correct localization of Talpid3 at centrioles is essential for its ability to promote centriole asymmetry and maturation (Fig. 8). The substitution mutation, C.2512C>T, is predicted to result in a truncated protein of 838 amino acids. This mutant protein correctly localized to the centrosome and was able to rescue centriole maturation but not CP110 removal or cilia formation. These data are consistent with our functional mapping of Talpid3 fragments (Figs. 8 and 2a). We conclude that the Talpid3 mutations found in JBTS and other lethal ciliopathies patients can be classified into three groups. One group of mutations resides in the amino-terminal portion (residues 1–465) of Talpid3, results in early termination and/or production of highly unstable proteins, and fails to support centriole maturation and ciliogenesis. The second group of mutations maps near the coil–coil domain (466–500 aa), disrupts the localization of Talpid3 to centrosomes, and abrogates centriole maturation and ciliogenesis. The third group of mutations partitions to the carboxy-terminal region (700–1533 aa) and results in truncation mutants that include residues 400–700, which enable normal centriole maturation but abrogate ciliogenesis. Although additional experiments are required, our data also suggest that diverse Talpid3 mutations, which result in defects at different stages of centriole maturation and ciliogenesis, could explain the spectrum of pathologies observed in JBTS and other lethal ciliopathies patients.

## Discussion

In this study, we identified Talpid3 and C2CD3 as critical regulators of DCP removal and revealed that removal of certain DCPs constitutes another level of control for DA assembly. We also revealed that a centriolar protein network, comprised of Talpid3, C2CD3, and OFD1, differentially regulates the assembly of other distal centriole structures, CS, and the actin network. Our studies could ultimately link defects in Talpid3 function to Joubert Syndrome and other lethal ciliopathies and explain the spectrum of pathologies associated with human mutations in OFD1 and C2CD3.

Until now, the controlled removal of DCPs and its relationship with DA assembly have not been methodically studied. This is due, in part, to the paucity of daughter centriole proteins identified thus far (three to-date). Indeed, it is intriguing that few daughter proteins have been identified, given that a substantially larger number of mother-specific proteins have been uncovered. We found that the removal of these three DCPs is an all-or-none event, such that future mother centrioles either remove or retain each of them (Fig. 1a and Supplementary Fig. 1a). This could be partially explained by the hierarchical recruitment/organization of DCPs (Fig. 3d). We found that control of CEP120 removal is critical for the removal of the other two DCPs. Since the CS pools of C2CD3 and OFD1 are not required for centriole maturation (Fig. 4a and Supplementary Fig. 5), and the centrosomal pools of Talpid3, C2CD3, and OFD1 reside exclusively at the distal ends of centrioles, our data imply that the removal of DCPs may be triggered or regulated at the distal end of the maturing mother centriole, where CEP120 and Centrobin co-localize with, or are juxtaposed to, Talpid3 and C2CD3. However, future studies will be required to determine how Talpid3 and C2CD3 regulate the removal of DCPs, given that the overall abundance of DCPs is not affected by the loss of Talpid3 or C2CD3 (Supplementary Fig. 4b). One possibility is that Talpid3 and C2CD3 enforce localized, centrosome-specific protein degradation of DCPs. It is also possible that Talpid3 and C2CD3 recruit an enzyme, such as a protein kinase, that alters the conformation and recruitment of DCPs. Also, since CEP120 and Centrobin interact with microtubules and regulate their stability[2,4],

the modifications/decorations of microtubules during centriole maturation could also affect the localization of DCPs. The possibility that Talpid3 and C2CD3 regulate such modifications cannot be excluded at this time.

We established that removal of DCPs constitutes another layer of DA assembly control based on the following evidence. First, defects in the removal of DCPs are accompanied by aberrant DA assembly in Talpid3 and C2CD3 KO cells (Figs. 1a and 4a) and, potentially, in patients harboring mutations in these genes. Both defects can be rescued by the same Talpid3 fragment, consisting of residues 400–700, in Talpid3$^{-/-}$ cells (Fig. 2a). Second, mimicking defective DCP removal observed in KO cells—by targeting CEP120 and Centrobin to both centrioles—suppressed DA assembly, and disruption of symmetrical localization of Centrobin in Talpid3$^{-/-}$ and C2CD3$^{-/-}$ cells reversed this inhibitory effect (Fig. 3a, b). Furthermore, we found that removal of DCPs is important for proper OFD1 localization, which in turn initiates DA assembly (Fig. 5). Therefore, our observations suggest how DCPs removal is mechanistically and temporally linked to subsequent maturation events such as appendage formation (Fig. 9a). How removal of DCPs promotes OFD1 recruitment remains to be determined. One possibility is that the inability to remove DCPs could sterically prevent the proper recruitment or localization of OFD1, and this mechanism is supported by the observations that PACT–CEP120 and PACT–Centrobin expression blocked OFD1 recruitment (Fig. 5a). It is also possible that removal of DCPs constitutes a checkpoint monitored by unknown regulatory protein(s) that also control the recruitment of OFD1 and initiation of DA assembly. Moreover, loss of DA has also been found recently in Talpid3$^{-/-}$ chicken cells[34], suggesting that the function of Talpid3 in mother centriole maturation is evolutionarily conserved. We previously found that knock-down of Talpid3 did not cause a centriole maturation defect[36], which we ascribe to the incomplete knock-down of Talpid3 afforded by RNAi, reinforcing the importance of using genetically null cells for studying centriolar protein function. Our work could thus explain certain genotype–phenotype relationships in ciliopathies, as particular patient alleles may reduce the abundance of Talpid3 protein on centrosomes, whereas other alleles—such as the ones shown here (Fig. 8)—may abolish Talpid3 protein expression altogether.

Our work may also shed further light on another important question, namely, how vesicle docking to basal bodies is intricately coupled to DA assembly. We speculate that Talpid3 is a bi-functional protein: whereas a middle fragment of the protein is required for organelle asymmetry and maturation as well as IFT recruitment, the carboxy-terminal half of Talpid3 could engage with determinants on early ciliary vesicles and initiate ciliogenesis (Fig. 9b). Our previous study showed that Talpid3 directly interacts with Rab8a and Rabin8, and this event could promote vesicle docking[36].

In this study, we have also unveiled both overlapping and distinct roles for Talpid3, C2CD3, and OFD1 in assembling distal-end structures, CS, and the actin network (Fig. 7), allowing us to study their interdependencies and functional relationships in DCP removal and DA assembly. Among these three proteins, Talpid3 plays a unique role in the organization of cytoplasmic actin, while C2CD3 is specifically required for SDA assembly. In terms of CS organization, Talpid3 antagonizes the function of C2CD3 and OFD1. Importantly, these data suggest that assembly of the SDA, CS, and actin network are independent of DCP removal and DA assembly. We also uncovered distinct roles for Talpid3, C2CD3, and OFD1 in assembling the distal end CEP350/FOP/CEP19 complex and demonstrated that the recruitment of CEP19, but not CEP350 or FOP, is linked to DCP removal and DA assembly (Figs. 7d and 9a). On the other hand, C2CD3 plays a unique role in regulating the recruitment of CEP350 and FOP. Considering

the proximity of CEP350 and FOP to SDA, these data also suggest C2CD3 as a key organizer of sub-distal structures.

Our results are interesting in light of recent findings on C2CD3 and OFD1, both of which traffic from CS to centrioles[25,46]. C2CD3 is conserved in worms (SAS-1)[50] and birds (Talpid2). Interestingly, the avian Talpid2 mutation results from a 120 amino acid carboxy-terminal deletion, which produces Hh and limb phenotypes also seen in Talpid3 mutants, and, interestingly, Talpid2 and Talpid3 animals exhibit cranio-facial defects[51]. C2CD3 is required for ciliogenesis, and human mutations in this gene lead to JBTS and OFD-type syndrome with additional features of JBTS[24,52,53], suggesting potentially overlapping disease mechanisms. Furthermore, SDA and DA assembly and CV recruitment are nearly abolished after depleting human C2CD3 and in Talpid2 mutant cells[24,51]. Moreover, C2cd3 and Ofd1 silencing leads to abnormally short or long centrioles[24,26], respectively, and Talpid3 ablation similarly promotes aberrant centriole elongation (Fig. 1b and refs. [34,36]). Given that all three genes (Talpid3, Ofd1, and C2cd3) have been implicated in ciliopathies and that several inter-related sets of defects result from the loss of each gene, these findings point to a potentially important and intimate relationship between Talpid3, OFD1, and C2CD3 in assembling an essential functional complex at basal body distal ends, defects in which lead to human ciliopathies.

Our work has revealed an extensive, overlapping set of functional similarities between Talpid3 and OFD1, which is also mutated in JBTS[54,55]. It will be important to understand the molecular basis for anomalies observed in JBTS patient cells by assessing phenotypes known to be associated with Talpid3 loss. It will also be interesting to determine whether disease-associated Ofd1 and C2CD3 mutations result in defects in recruitment or localization of Talpid3, DCP removal, basal body maturation, Rab trafficking, and ciliogenesis. JBTS patients with Talpid3 mutations show brain defects, although they exhibit varying degrees of pathology. In this study, we identified three types of Talpid3 mutations that affect different stages of centriole maturation and ciliogenesis, suggesting a potential causative link between defects in these events and varying degrees of pathology in JBTS patients. Future experiments will determine whether disease mutations map to distinct fragments with specific functions that we have identified herein, thereby producing unique disease phenotypes of differing severity.

## Methods

**Cell culture and gene-editing using CRISPR/Cas9.** Human retinal pigment epithelial (RPE1-hTERT) and human embryonic kidney (HEK293T) cells were obtained from ATCC. Cells were grown in DMEM supplemented with 10% FBS. To induce cilia formation, RPE1 were incubated in DMEM without FBS for 24 or 48 h. To generate CRISPR KO cells, RPE1 cells were infected with lentivirus expressing Flag-Cas9 and sgRNA and grown for 10 days, after which the cells were examined by IF and separated as single cells into 96-well plates. After 2 weeks, the colonies were analyzed for genome editing. sgRNAs used included: sgCTL (5′-GAGACGTCTAGCA CGTCTCT-3′), sgTalpid3 (5′-GATGATGTTCTTCATGACCT-3′), sgC2CD3 (5′-GGAGGAGGTGATCTTCAATG-3′), sgOFD1 (5′-GGTGCTTGTGAATTCTT TCA-3′), and sgCEP128 (5′-GCTGCCAGATCAACGCACAGGG-3′).

**Transfection and lentivirus infection.** Polyethylenimine (PEI) was used for plasmid transfection in 293T cells. DNA and PEI (1 mg/ml) were added at 1:5 to 1:8 ratio. Lentiviral supernatant was prepared by co-transfection of the lentiviral plasmid with Δ8.2 envelope and VsVG packaging plasmids into 293T cells using PEI. Lentivirus supernatants were harvested 48–72 h post-transfection. RPE1 cells were incubated with virus supernatants in the presence of 8 µg/ml polybrene for 6–10 h, and medium was changed thereafter. siRNAs were transfected into RPE1 cells using RNAiMAX (Invitrogen, Carlsbad, CA) according to the manufacturer's protocol. siRNAs were synthesized by Dharmacon with the following sequences: non-specific control (5′-AATTCTCCGAACGTGTCACGT-3′), Centrobin (5′-GGATGGTTCT AAGCATATC-3′), CEP120 (5′-GAUGAGAACGGGUGUGUAU-3′, 5′-AAACCG AGCGACAAGAAUU-3′, and 5′-GGAUUUAAGAACCGCUCAA-3′) siRNAs, and the siRNA pool against CEP83 (L-021034–02–0005).

**DNA constructs**. To generate Myc-tagged Talpid3 proteins, human Talpid3 truncations were amplified by PCR and sub-cloned into the PCDH-Myc-Neo vector. Talpid3 mutants were generated by site-directed mutagenesis. The expression of Talpid3 truncations and mutants was confirmed by immunoblotting using a mixture of Talpid3_1 and Talpid3_2 antibodies or anti-Myc antibody. To generate Myc-tagged C2CD3 protein, a human C2CD3 cDNA was obtained from Kazusa DNA Research Institute (Kazusa-kamatari, Chiba, Japan) and cloned into PCDH-Myc-Neo vector. A plasmid expressing Myc-tagged OFD1 was obtained from Andrew M. Fry (University of Leicester, Leicester, UK). EGFP-SmoM2 was a gift from J.F. Reiter (University of California, San Francisco, USA). To generate Myc-tagged PACT–CEP120 and PACT–Centrobin, the PACT domain, CEP120, and Centrobin were amplified by PCR and sub-cloned into Plvx-Myc vector. Myc-tagged PACT–Neurl4 was generated by sub-cloning the PACT domain and Neurl4 into the PCDH-Myc-Neo vector. To generate Myc-tagged Talpid3Nter-C2CD3 and Talpid3Nter-OFD1, Talpid3Nter, C2CD3, and OFD1 were amplified by PCR and sub-cloned into PCDH-Myc-Neo vector. All PCR reactions were performed using high fidelity PfuTurbo DNA polymerases (Agilent), and the PCR-generated plasmids were further verified by DNA sequencing.

**Immunoprecipitation**. 293T or RPE1 cells were lysed in ELB buffer (50 mM Hepes pH 7, 150 mM NaCl, 5 mM EDTA pH 8, 0.1% NP-40, 1 mM DTT, 0.5 mM AEBSF, 2 μg/ml leupeptin, 2 μg/ml aprotinin, 10 mM NaF, 50 mM β-glycerophosphate, and 10% glycerol) on ice for 10 min, lysates were centrifuged at 16,000×$g$ for 15 min, and supernatants were incubated with 2 μg anti-Myc antibody (sc-40, Santa Cruz) and 15 μl Protein G Sepharose (17-0618-01, GE Healthcare) or 15 μl Flag beads (A2220, Sigma-Aldrich). For immunoprecipitation, 2 mg of the resulting supernatant was immunoprecipitated, and beads were washed with ELB buffer and analyzed by immunoblotting. Protein band intensities were quantified using Image J software. The uncropped blots are shown in Supplementary Fig. 6.

**Immunofluorescence microscopy**. Cells were fixed with cold methanol for 10 min or with 10% formalin solution (Sigma-Aldrich) for 15 min and permeabilized with 0.3% Triton X-100/PBS for 10 min. Slides were blocked with 3% BSA in PBS before incubation with primary antibodies. Secondary antibodies used were Cy3-conjugated (Jackson ImmunoResearch Laboratories, Inc.), Alexa Fluor 488-conjugated or Alexa Fluor 647-conjugated (Invitrogen) donkey anti-mouse, anti-rabbit, or anti-goat IgG. Cells were stained with DAPI, and slides were mounted, observed, and photographed using a microscope (63× or 100×, NA 1.4; Axiovert 200M, Carl Zeiss) equipped with a cooled CCD (Retiga 2000R; QImaging) and MetaMorph Software (Molecular Devices). Alternatively, an LSM 800 confocal microscope (63×, NA 1.4 Carl Zeiss) with Zen software (Carl Zeiss) was used. Super-resolution microscopy was performed using a structured-illumination microscopy (SIM) system (DeltaVision OMX 3D; Applied Precision). For SIM, a 100×, 1.4 NA oil objective (Olympus) was used with 405, 488, and 593 nm laser illumination and standard excitation and emission filter sets. 125-nm z-steps were applied to acquire raw images, which were reconstructed in 3D using SoftWoRx software (Applied Precision). Image analysis was performed using Photoshop (Adobe). Intensity of CS and DA proteins was quantified by Image J. Briefly, regions of interest were defined by drawing a circle (radius of 2.5 μm for PCM1 and 0.8 μm for CEP83 and CEP164) centered on the centrosome. Background values were measured from the same-sized circle in an adjacent region. Staining was analyzed in G0/G1 phase cells (serum-starved for 24 or 48 h) to achieve uniformity and to avoid oscillations in abundance during the cell cycle.

**Transmission electron microscopy (TEM)**. RPE1 cells were washed with PBS followed by fixation with 0.1 M sodium cacodylate buffer (pH 7.4) supplemented with 2% paraformaldehyde, 2.5% glutaraldehyde, and 0.1% Ruthenium red. Cells were post-fixed with 1% osmium tetroxide for 1.5 h at room temperature, and stained with 1% uranyl acetate, processed in a standard manner, and embedded in EMbed 812 (Electron Microscopy Sciences) for TEM. Serial thin (60 nm) sections were cut, mounted on 200 mesh or slotted copper grids, and stained with uranyl acetate and lead citrate. Stained grids were examined using an electron microscope (model CM-12; Philips/FEI) and photographed with a 4-k × 2.7-k digital camera (Gatan, Inc.).

**Antibodies**. Antibodies used include: Talpid3_1 (antigen:1–180, 1:500 for WB), Talpid3_2 (antigen: 847–1026, 1:500 for IF and WB), centrin (1:2500 for IF, 04-1624; Millipore), Rabbit anti-Flag (1:2000 for WB, F7425, Sigma), mouse anti-Flag (1:2000 for WB and 1:500 for IF, F1804, Sigma), goat anti-GFP (1:500 for IF, ab545025, Abcam), mouse anti-α-tubulin (1:5000 for WB, T5168, Sigma), goat anti-γ-tubulin (1:500 for IF, sc-7396, Santa Cruz), mouse anti-polyglutamylated tubulin (GT335) (1:2500 for IF, AG-20B-0020-C100, Adipogen), rabbit anti-IFT88 (1:1000 for WB and 1:500 for IF, 13967-1-AP, Proteintech), rabbit anti-IFT140 (1:200 for IF, 17460-1-AP, Proteintech), rabbit anti-Arl13b (1:2000 for WB and 1:2000 for IF, 17711-1-AP, Proteintech), rabbit anti-Rab8 (1:200 for IF, gift from J. Peränen), rabbit anti-TTBK2 (1:500 for IF, HPA018113, sigma), rabbit anti-CP110 (1:1000 for WB and 1:200 for IF), rabbit anti-OFD1 (1:2000 for WB and 1:500 for IF, gift of J.F. Reiter), rabbit anti-OFD1-2 (1:1000 for WB and 1:250 for IF, gift of A. Fry), mouse anti-CEP170 (1:500 for IF, 72-413-1, Invitrogen), rabbit anti-Neurl4 (1:500 for IF),

mouse anti-Centrobin (1:1000 for IF, ab70448, Abcam), rabbit anti-CEP120 (1:5000 for IF, gift from LH Tsai), rabbit anti-FBF1 (1:500 for IF, 11531-1-AP, Proteintech), rabbit anti-CEP83 (1:500 for IF, HPA038161, Sigma), rabbit anti-C2CD3 (1:500 for IF, HPA038552, Sigma), rabbit anti-CEP89 (1:50 for IF, gift from M. Bornens), rabbit anti-CEP164 (1:500 for IF, 45330002, Novus), rabbit anti-CEP19 (1:2000 for IF, ab74989, Abcam), rabbit anti-FOP (1:2000 for IF, A301-860A, Bethyl), rabbit anti-CEP350 (1:1000 for IF, NB100-59811, NOVUS), rabbit anti-ODF2 (1:100 for IF, H00004957-M01, NOVUS), rabbit anti CEP128 (1:2000 for IF; A303-348, Bethyl), and mouse anti-Centriolin (1:200 for IF; sc-365521, Santa Cruz).

**Statistics and reproducibility**. The statistical significance of the difference between two means was determined using a two-tailed unpaired Student's $t$-test. All data are presented as mean ± SD as specified in the figure legends. Differences were considered significant when $p < 0.05$. Results reported are from 2–3 independent biological replicates as noted in legends with reproducible findings each time. For all experiments, except as noted, $N \geq 100$ cells per sample were counted in three biologically independent experiments.

## Data availability
The uncropped western blots are shown in Supplementary Fig. 6. The data that support the findings of this study are available from the corresponding author (B.D.D.) upon reasonable request.

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

## Acknowledgements

We thank M. Owa for helpful comments on our manuscript. We thank M. Bornens, A. Fry, and J. Reiter for plasmids and antibodies detailed in the Methods section. We are especially grateful to T. Kanie and P. Jackson for *CEP350* and *FOP* null RPE1 cells. We thank F. Liang and the NYU School of Medicine Imaging core for assistance with EM. The Rockefeller University Bio-Imaging Center provided assistance. The project described was supported by Award Number S10RR031855 from the National Center for Research Resources. The content is solely the responsibility of the authors and does not necessarily represent the official views of the National Center for Research Resources or the National Institutes of Health. Work in BDD's laboratory was supported by 9R01GM120776-05A1 and a DOD prostate cancer postdoctoral training award W81XWH-16-1-0392 to L.W.

## Author contributions

L.W. and B.D.D. designed the experiments, and L.W. conducted the experiments. M.F. generated *Cep128*<sup>−/−</sup> cell line and contributed to the SIM experiments. W.F. contributed to the CRISPR knock-out experiments and EM studies. L.W. and B.D.D. analyzed the data and wrote the paper, and all authors were involved in reading and correcting the manuscript.

## Additional information

**Competing interests:** The authors declare no competing interests.

