## [Peer Review File · Nature Communications]

Reviewers' comments:

Reviewer #1 (Remarks to the Author):

In this study, the authors identified Talpid3 and C2CD3 as regulators to generate asymmetry of centrioles by removing daughter centriole-enriched proteins (DCPs). The authors also show these two proteins, together with OFD1, function in a network but differentially regulate assembly of the distal appendage, the Cep350/FOP/Cep19 module, the centriolar satellite, as well as the actin network. They finally show hierarchy and flow chart of those proteins and multiple domain functions of Talpid3. Main approaches used in this study are based on combinations of CRISPR-Cas9 knockout, siRNA knockdown and rescue, and immunofluorescence. Overall, while I appreciate the quantity of the work in this paper and the claims on molecular networks between DA/SDA and DCP proteins are potentially interesting to the field, I have several concerns that would preclude publication in Nature Communications at this stage.

One issue before discussing scientific contents is that the structure of the manuscript appears to be in disorder. Particularly, the main figures contain too many pieces of data while only two supplemental figures are included, which looks quite unbalanced. For example, to figure out what are on each picture/graph in Fig 1, I had to super-magnify the figure. In addition, for example, Fig 1A contains a ton of pictures and graphs. I was confused to know which one is mentioned when the text merely refers to like "Fig 1A". This is also the case for many places on other figures. By splitting them and putting part into supplementary figures, the figures should be more organized so that readers can easily follow them.

The most important scientific concern is the way the authors quantify the data. Most of data are shown as like "% cells with XX" but the signal intensity should be measured, at least for backup, instead in many places to solidify the conclusions. The current criteria for just counting the number of dots at centrioles/centrosomes seem obscure to me, what is the border between "positive" and "negative"? The super-resolution observations in Figure 5 are intriguing, but the relative position of the proteins needs to be clarified by quantifying the diameter of rings and distance from reference markers.

Specific comments are as follows.

1. Generally speaking, 46 references just in Introduction are too many. It is OK if appropriate, but the authors should check the references once again and organize them if necessary.
2. The order of figures appearing in the text is mixed up. For example, Fig. S2B comes earlier than Fig S1 (page 6), Figs 5 and 6 suddenly come after Figs 1 and 2 in pages 7 and 10, respectively. This is so confusing and makes the manuscript look less logical.
3. Page 7, line 9, the authors state "Since centriole duplication is grossly normal in Talpid3^{-/-} cells (Fig. S1A)", but Fig S1A only shows representative pictures. Quantification of the centriole numbers should be shown as well.
4. What is the cause of centriole elongation phenotype by Talpid3 depletion? Would it be possible to rescue the phenotype by depletion of DCP proteins?
5. In pictures in Fig 2A, CEP164/CEP83 signals reduced but still remained in PACT-CEP120/Centrobins-treated cells. In such case, signal intensity should be measured as in Fig 6B, instead of counting "cells with XX". I am concerned how the authors define "cells with" and "without" in situations like this throughout the study. Even worse, the background of the PACT-CEP120-CEP164 picture in Fig 2A is obviously lower than that of control, making me worried that contrast was inappropriately adjusted. Representative pictures with the same contrast and quantification of the signal intensity should be shown in Fig 2A and similarly in many of other figures.
6. If I am not mistaken, there seems to be a discrepancy in Fig. 2A. How can only Centrobins rescue the DA assembly in Talpid3 KO cells, even if Centrobins recruitment at centrioles is regulated by Cep120? I would expect a similar rescue by Cep120 or Neurl4 depletion.
7. In the Western blotting in Fig 2B, the Centrobins expression level in siCentrobins-treated Talpid3^{-/-} cells looks comparable to that in siControl. Did siCentrobins surely work? Do the authors have

any other information on this matter?

siCEP120 data in Fig 2B is a typical one that should be moved to supplemental figures because it is minor and negative.

8. In page 11, line 14, the authors mention "We also found that Talpid3 could interact with C2CD3 through residues 466-1533 (Fig. 3D)". Here only the residues 466-1533 is mentioned although Fig 3D shows multiple truncated fragments and those other than the 466-1533 are positive for IP. Please add more detailed description for the data. Furthermore, in figures 3 and S1, Co-IP experiments between Talpid3 and C2CD3 seem not to be consistent in terms of responsible regions and stoichiometry. The quantification of the bands should be provided for each lane.

9. Page 10, 4th line from the bottom, needs references for "consistent with previous reports".

10. Fig 4D was not mentioned in the text.

11. Please specify what the "high resolution microscopy" in page 14, line 2 is. The figure legend of Fig 5 says this is the Airyscan but when another method than the previous one (SIM) was used, it is important to mention that in the text.

12. In Fig 5B, it is hard to tell what are going on with this resolution and to support the claim here. Could those be done by SIM?

13. In Fig 6B, the signal intensity of PCM1 in Talpid3^{-/-} looks much higher than that of control when looking at the pictures. However, in the graph, the difference is not that large (less than 2-fold increase). In addition, there is almost no PCM1 signal in control. I am concerned about how the authors quantified these data. Although the method is described, my concern is how much accurately the quantification was indeed done.

14. In Fig 6D, what is the difference between + and ++? Please define them.

15. Make notation consistent between the text and the figures. For example, Cep83 vs CEP83. Also check the notation of siRNA. I think notation like "siCEP120" is more common compared to "Si CEP120" as in the figures. This is also the case for "sgRNA".

16. Typos. Page 9, 2nd line from the bottom; though -> through.

Page 13, last 2 sentences; check the articles and the tense ("the DC" and "co-localized").

Reviewer #2 (Remarks to the Author):

In this manuscript Wang et al. characterize the role of Talpid3 in the regulation of asymmetric localization of daughter centriole specific-proteins and centriole maturation. Talpid3 knockout RPE-1 cells are defective in the localization of distal appendage proteins, docking of the ciliary vesicles and ciliogenesis. By generating a series of Talpid3 truncations, the authors mapped the region required for centriole maturation, in addition to the region responsible for ciliogenesis initiation.

The authors described a function for the Talpid3-c2cd3-Ofd1 complex in the maturation of centrioles and in the asymmetric localization of daughter centriole specific-proteins. The authors also investigate the localization and function of ciliopathy-associated mutants of Talpid3 and examine their defects in localization and/or centriole-association functions.

Overall, the experiments are well performed and the manuscript is well written. Results of this manuscript will help our current understanding of centriole maturation and the role of daughter centriole specific-proteins in the regulation of distal appendages assembly and ciliogenesis. However, there are some concerns with some experiments and parts of the manuscript, and addressing them is essential for the clarity of the results.

Main comments:

1- Talpid3 knockout cells are defective in the removal of CP110 from the mother centrioles in response to serum starvation. CP110 suppresses centriole elongation by capping the distal end of

the centrioles [1, 2]. However, centrioles are abnormally elongated in Talpid3 knockouts. Can the authors explain the mechanism of centriole elongation in the presence of CP110? Furthermore, in the previous publication investigating the role of Talpid3 using RNAi [3], the authors did not report any defects to the centriole length which contradicts the present data. This need to be clarified.

2- The first section of the results is concluded by the statement that "the assembly of distal appendages is sufficient for recruitment of the IFT machinery but not for initiation of vesicle docking or subsequent events during ciliogenesis". However, in the following paragraph they state that "all subsequent defects observed in Talpid3^{-/-} cells, including the failure to dock vesicles, TTBK2 recruitment, CP110 removal, and IFT transport could be attributed to the DA assembly defect", which apparently contradicts their interpretation of their results.

3- The authors attribute functions of several regions of Talpid3 by generating truncation constructs. It would be important to describe the known domains of Talpid3 and explain why they chose these fragments. Furthermore, I suggest using the word region or fragment instead of "domain" since the known domains of Talpid3 are not described in the manuscript.

4- On page 7, the authors report that they identified a region encompassing amino acids 466-700 of Talpid3 which is required for asymmetric localization of Centrobin and Cep120. However, the data for this statement does not appear to be provided in Figure 1D.

5- The authors perform rescue experiments to identify the minimal regions of Talpid3 responsible for the functions reported in the manuscript. However, the major phenotype of regulating the asymmetric localization of daughter centriole specific-proteins, namely CEP120 and Centrobin, is not investigated and should.

6- On page 10, the authors claim that the centriolar localization of Talpid3 is essential for its interaction with CEP120 (supp. figure 1). However, in this figure C.1697A>T mutant also interacts with Flag-Cep120, albeit much weaker than 466-1000aa fragment.

7- The authors had previously reported the accumulation of centriolar satellite proteins PCM1, BBS4 and CEP290 in Talpid3-depleted cells [3]. However, to examine the role of centriolar satellites in centriole maturation in the absence of Talpid3, they used a PCM1 knockout cell line which completely lacks centriolar satellites. It may be more relevant to investigate the role of centriolar satellites in centriole maturation using a cell line which displays the accumulation of centriolar satellites, similar to Talpid3^{-/-} cells.

8- In Figure 3E, the results of the transfection of N-terminal region of Talpid3 alone should be included. Moreover, the statement that "to some extent, the centriolar recruitment of OFD1 by Talpid3 plays an important role in the regulation of asymmetric localization of daughter centriole proteins" is not totally correct, since they showed that OFD1 appears dispensable for this phenomenon (Figure 3A&B).

9- In Figure 3E, the authors should present the results of transfecting cells with the N-terminal domain of Talpid 3 only, not fused to other proteins.

Minor points:

1- On page 6, the data on the accumulation of centriolar satellites in the absence of Talpid3 is presented in a previous publication, not in figure 1A. Please clarify this in the manuscript.

2- Please provide the quantification of the data presented in supp. Figure 1 for centriole duplication.

3- FACS data in supp. Figure 2B. is pertaining to 3 cell lines including Talpid3 KOs. Please include this in the figure legend.

References:

1. Schmidt, T.I., Kleylein-Sohn, J., Westendorf, J., Le Clech, M., Lavoie, S.B., Stierhof, Y.D., and Nigg, E.A. (2009). Control of centriole length by CPAP and CP110. *Curr Biol* 19, 1005-1011.
2. Spektor, A., Tsang, W.Y., Khoo, D., and Dynlacht, B.D. (2007). Cep97 and CP110 Suppress a Cilia Assembly Program. *Cell* 130, 678-690.
3. Kobayashi, T., Kim, S., Lin, Y.-C., Inoue, T., and Dynlacht, B.D. (2014). The CP110-interacting proteins Talpid3 and Cep290 play overlapping and distinct roles in cilia assembly. *The Journal of Cell Biology* 204, 215.

Reviewer #3 (Remarks to the Author):

Centrosomes/centrioles are structurally complex organelles comprised of a number of distinct

structural elements each of which contains specific protein components. During centrosome biogenesis, many of these components display specific spatiotemporal recruitment or localization patterns. For example, newborn centrioles carry Talpid3, C2cd3 and Ofd1 at their distal end, whereas appendage proteins, which also localize to the distal end, are recruited at much later stages in a process called centriole maturation. Even more mysterious is a group of proteins called daughter centriole enriched proteins (DCPs) including centrobins, cep120 and nerul4; they are present at newborn centrioles but are somehow removed prior to the onset of centriole maturation. Neither the mechanism nor the significance of DCP removal is understood. The manuscript by Wang et al presents a large body of data for an intriguing proposal that DCP removal is regulated by Talpid3 and C2cd3, and that the removal is required for Ofd1 recruitment and subsequent appendage assembly both essential for ciliogenesis.

I find that the author's conclusion, if confirmed, is very novel and significant, and will greatly advance our knowledge on centriole and cilia biogenesis. Given its potential impact, I think the authors should carefully address the following concerns, which I believe will strengthen the data for their exciting/provocative proposal while resolving some inconsistency.

Major concerns:

a). DCP REMOVAL IS REQUIRED FOR DISTAL APPENDAGE (DA) ASSEMBLY (Fig. 2 & 4):

Some of the most important experiments presented by the authors to support this key conclusion are described in Fig 2 & 4, where they show that knockdown of the DCP (centrobin) can bypass the requirement of Talpid3 or C2cd3 for DA assembly (i.e. able to rescue DA assembly in Talpid3^{-/-} or C2cd3^{-/-} cells). While the data is very intriguing, as its current form, it only means that DA proteins can "gather around" the centrosome in Talpid3^{-/-} or C2cd3^{-/-} cells upon centrobin knockdown, which is not the same as the actual, physical assembly of the DA. This issue is a very critical as it is known that the distribution of many centrosomal proteins in the cell or around the centrosome can be altered or impacted by the pericentriolar satellites, a structure known to associate with many centrosomal proteins including those being studied here (Talpid3, C2cd3, Ofd1...). The authors should use 3D-SIM to confirm the rescue of DA assembly in both Talpid3^{-/-} and C2cd3^{-/-} cells by showing that these DA components are indeed assembled into a ring-like structure at the right location (distal end of the centriole) and size (~500 nm). Non-specific accumulation of DA proteins around the centrosome should not lead to such specific patterns.

Similar experiments should be done for the Ofd1 rescue data: Does centrobin knockdown in C2cd3^{-/-} cells enable Ofd1 to localize to the right location (centriole distal end) with the right pattern (small ring) or merely induce non-specific accumulation of Ofd1 around the centrosome?

Similar experiments should be done for the CEP19 rescue data: true rescue or non-specific accumulation (Fig. 1D & 6A).

b). Ofd1 is known to be loaded to the distal end of pro-centrioles from the beginning of centriole biogenesis, and stay there throughout the life of the centriole. Ofd1 is also known to localize to the pericentriolar satellites, which normally accumulate around the centrosome/MTOC. In Fig 4, the authors reported that DCP removal is required for the recruitment of "centrosomal Ofd1". What are these centrosomal Ofd1 molecules? Are they those loaded during pro-centriole formation or those associating with the satellites or both? Does PACT-centrobin or PACT-cep120 block the loading of Ofd1 onto pro-centrioles, satellites or both (Fig. 4A)?

c) I wonder if the authors know where in the centrosome PACT-centrobin or PACT-cep120 is localized to, as both fusion constructs can associate with the PCM of the mother centriole via the PACT domain, and with the centriole (including pro-centrioles that do not have the PCM) via the domains residing in cep120 or centrobin. In this sense, which population of PACT-centrobin or PACT-cep120 inhibits the recruitment/loading of Ofd1 (Fig. 4A)? Like the issue raised in (b), it can lead to completely different interpretations.

d). Other issues regarding localization studies: In some images, the authors used GT335 to mark individual centrioles (e.g. Fig. 3C), but in others (e.g. Fig. 2D), gamma-tubulin was chosen to mark individual centrosomes each of which may carry one (G1) or two (S/G2) centrioles. The interpretation of the localization data will be very different in each case. To fully reveal the nature of the sequential recruitment and removal of different DCPs in WT or mutant cells, the authors must use centriolar marker such as centrin or GT335 (instead of gamma-tubulin) to separately

visualize G1 and S/G2 centrioles. Otherwise, I have hard time to understand some of the data presented here, especially the data for Ofd1 loading (see (b)).

e). From the EM study shown in Fig 1B, about 40% of centrioles in Talpid3^{-/-} cells are significantly elongated (> 1µm). However, of all the IF images collected from Talpid3^{-/-} cells (e.g. Fig. 1A & E, Fig. 2B, Fig. 3C), none of the centrioles shown appear to be elongated (by GT335 staining). Why is that?

f) Talpid3 and C2cd3 can regulate the property of the centriole distal end and centriolar satellites. In the deletion analysis shown in Fig. 1D, can the author address which activity of Talpid3 and C2cd3 is responsible for DCP removal? How can the authors be sure that it is the distal end associated activity that regulates DCP removal?

g). Personally, I find this paper not very easy to read. The amount of data is huge, supporting a very long story that is divided into parts not intuitively connected, lacking a primary focus. It will probably work better if the authors choose one primary finding as the focus and go deeper to reveal its significance. For example, for me, the connection between Talpid3/C2cd3/Ofd1 (centriolar satellites), DCP removal and DA assembly is very significant. I would love to see more detailed, focused analyses on this part only.

Reviewer 1

One issue before discussing scientific contents is that the structure of the manuscript appears to be in disorder. Particularly, the main figures contain too many pieces of data while only two supplemental figures are included, which looks quite unbalanced. For example, to figure out what are on each picture/graph in Fig 1, I had to super-magnify the figure. In addition, for example, Fig 1A contains a ton of pictures and graphs. I was confused to know which one is mentioned when the text merely refers to like “Fig 1A”. This is also the case for many places on other figures. By splitting them and putting part into supplementary figures, the figures should be more organized so that readers can easily follow them.

We are grateful to this Reviewer for his/her generally enthusiastic comments. We thank the Reviewer for this advice, and we agree that this suggestion has improved the organization of our manuscript. Briefly, we now have 6 supplementary figures. Fig. 1 was split into two figures (now Figs. 1 and 2). In addition, certain data in the main figures were moved to a supplementary figure as requested in point 7 below.

The most important scientific concern is the way the authors quantify the data. Most of data are shown as like “% cells with XX” but the signal intensity should be measured, at least for backup, instead in many places to solidify the conclusions. The current criteria for just counting the number of dots at centrioles/centrosomes seem obscure to me, what is the border between “positive” and “negative”?

To solidify our data, we have now extensively quantified the intensity of centrosomal markers in original Fig. 2A and 2B (now Fig. 3a and 3b) and other proteins here and as requested in point 5 below. Please also see the data in Fig. S2A for further support. As suggested, we have also confirmed our primary findings by measuring the diameter of rings associated with distal appendages (DA) as well as Cep19 and OFD1 staining in the key rescue experiments of *Talpid3*^{-/-} and *C2CD3*^{-/-} cells using 3D-SIM. Please see new Figs. 6 and 7 and Figs. S1C, S2A and C, and S5C.

The super-resolution observations in Figure 5 are intriguing, but the relative position of the proteins needs to be clarified by quantifying the diameter of rings and distance from reference markers.

As discussed above, please see our measurements of ring diameters in Figure 5 (now Figs. 6), as well as Figs. S1, S2, and S5.

1. Generally speaking, 46 references just in Introduction are too many. It is OK if appropriate, but the authors should check the references once again and organize them if necessary.

We agree there are many references in the Introduction. This primarily stemmed from the fact that there are few, if any, reviews that broadly discuss distal and sub-distal appendages and the diseases caused by mutations in centriolar proteins we have studied in this paper. We feel that a rigorous and scholarly discussion of this topic requires inclusion of many of these references. Nevertheless, we have shortened the reference list for this section by removing references for the less germane papers.

2. The order of figures appearing in the text is mixed up. For example, Fig. S2B comes earlier than Fig S1 (page 6), Figs 5 and 6 suddenly come after Figs 1 and 2 in pages 7 and 10, respectively. This is so confusing and makes the manuscript look less logical.

As requested, we have modified the sequence of supplementary figures. They are now mentioned in order, as they appear in the text.

3. Page 7, line 9, the authors state “Since centriole duplication is grossly normal in *Talpid3*^{-/-} cells (Fig. S1A)”, but Fig S1A only shows representative pictures.

We thank the Reviewer for pointing this out. We have now included the quantification in Fig. S1A.

4. What is the cause of centriole elongation phenotype by *Talpid3* depletion? Would it be possible to rescue the phenotype by depletion of DCP proteins?

Our primary reason for including the elongated centriole phenotype in Fig. 1 was to show that the *Talpid3* knock-out recapitulates the phenotypes observed after siRNA treatment. Importantly, we found that failure to remove DC proteins does not strictly correlate with aberrantly elongated centrioles, since centrioles are elongated in *Talpid3*^{-/-} cells, whereas they are shortened in *C2CD3*^{-/-} cells. We have now mentioned this in the text (p. 15). Centriole length control by *Talpid3/C2CD3/OFD1* is a very interesting observation. However, due to the focus of this paper, space limitations, and the length of time allotted to revising this manuscript, we propose to investigate this issue in future studies outside the scope of this manuscript.

5. In pictures in Fig 2A, CEP164/CEP83 signals reduced but still remained in PACT-CEP120/Centrobin-treated cells. In such case, signal intensity should be measured as in Fig 6B, instead of counting “cells with XX”. I am concerned how the authors define “cells with” and “without” in situations like this throughout the study. Even worse, the background of the PACT-CEP120-CEP164 picture in Fig 2A is obviously lower than that of control, making me worried that contrast was inappropriately adjusted. Representative pictures with the same contrast and quantification of the signal intensity should be shown in Fig 2A and similarly in many of other figures.

Please see our quantification of signal intensity for original Figs. 2A and 2B (now Fig. 3a and 3b) in Fig. S2A. For the Cep164 channel in Fig 2A (new Fig. 3a), we used the same exposure time and contrast settings for all of the samples. The background difference could be due to slight differences in the number of z-stacks or sample variation. To minimize the background differences, we updated new Fig. 3a by including the exact same number of z-stacks for each sample.

6. If I am not mistaken, there seems to be a discrepancy in Fig. 2A. How can only Centrobin rescue the DA assembly in Talpid3 KO cells, even if Centrobin recruitment at centrioles is regulated by Cep120? I would expect a similar rescue by Cep120 or Neurl4 depletion.

We do not believe there is a discrepancy here. We speculate that during the maturation of daughter centrioles, removal of DCPs at the G1/S transition could be a prerequisite for the acquisition of appendages occurring later in G2 phase. To test this hypothesis, DCPs were forced to symmetrically localize on both centrioles in wild-type RPE1 cells by expressing fusion proteins containing the PACT domain. We found that PACT-CEP120 and PACT-Centrobin, but not PACT-Neurl4, were able to prevent the localization of OFD1 and DA proteins, CEP83 and CEP164 (now Figs. 3A and 5A). To further examine whether the failure to remove CEP120 and Centrobin inhibits DA assembly, we knocked down Centrobin in Talpid3^{-/-} cells using siRNAs. In agreement with our hypothesis, we found that depletion of Centrobin was able to substantially rescue the assembly of DA (now Fig. 3B). Knocking down Cep120 blocked centriole duplication and resulted in cells with one centriole or no centrioles (now Fig. 3B), as reported in a previous study¹, preventing us from confirming its role in inhibiting DA assembly. We did not knock down Neurl4 in Talpid3 KO cells because the data in Fig. 2A (now Fig. 3A) suggested that removal of Neurl4 is not required for DA assembly. Also, previous studies have shown that Cep120, Centrobin, and Neurl4 are not functionally identical¹⁻⁴, which is in line with our data here that Cep120, Centrobin and Neurl4 play different roles in centriole maturation.

7. In the Western blotting in Fig 2B, the Centrobin expression level in siCentrobin-treated Talpid3^{-/-} cells looks comparable to that in siControl. Did siCentrobin surely work? Do the authors have any other information on this matter?

We have repeated the western blotting and updated the figure. We also present immunofluorescence data in new Fig. 3D to confirm that siCentrobin indeed works well.

siCEP120 data in Fig 2B is a typical one that should be moved to supplemental figures because it is minor and negative.

We have removed the IF data from this figure and moved the western blot data to Fig S2B. We believe that all other data should remain in the main figures but are happy to move any other data to the Supplement at the discretion of Reviewers and the Editor.

8. In page 11, line 14, the authors mention "We also found that Talpid3 could interact with C2CD3 through residues 466-1533 (Fig. 3D)". Here only the residues 466-1533 is mentioned although Fig 3D shows multiple truncated fragments and those other than the 466-1533 are positive for IP. Please add more detailed description for the data. Furthermore, in figures 3 and S1, Co-IP experiments between Talpid3 and C2CD3 seem not to be consistent in terms of responsible regions and stoichiometry. The quantification of the bands should be provided for

each lane.

As requested by the Reviewer, the intensities of these bands were quantified as described in the figure legend, and the results are indicated below each panel. We have also augmented discussion of these results in the Results section.

9. Page 10, 4th line from the bottom, needs references for “consistent with previous reports”.

The references have been added.

10. Fig 4D was not mentioned in the text.

We thank the Reviewer for pointing this out. We have now mentioned this figure (now Fig. 5D) at the end of the section entitled “Asymmetric localization of DC enriched proteins is required for proper localization of OFD1” (p. 14).

11. Please specify what the “high resolution microscopy” in page 14, line 2 is. The figure legend of Fig 5 says this is the Airyscan but when another method than the previous one (SIM) was used, it is important to mention that in the text.

In response to comments below (point 12) and from Reviewer 3, we have now re-evaluated these data with a higher resolution approach, namely, SIM. We have therefore replaced all of the Airyscan data with SIM data. The description of “high resolution microscopy” was modified to “SIM”.

12. In Fig 5B, it is hard to tell what are going on with this resolution and to support the claim here. Could those be done by SIM?

As described in point 11, we have now replaced all these Airyscan data with SIM data.

13. In Fig 6B, the signal intensity of PCM1 in Talpid3^{-/-} looks much higher than that of control when looking at the pictures. However, in the graph, the difference is not that large (less than 2-fold increase). In addition, there is almost no PCM1 signal in control. I am concerned about how the authors quantified these data. Although the method is described, my concern is how much accurately the quantification was indeed done.

Centriolar satellites (CS) are granules scattered around the centrosome, rendering it challenging to quantify their intensity precisely. As a result, following recent studies⁵⁻⁷, we quantified CS intensity by drawing a circle (radius=2.5 μm) around the centrosome to cover the majority of CS granules. We also quantified CS intensity by drawing smaller circles (radius=1.25 μm) for new Fig. 7b, which led to the same conclusion (please see Fig. R1 below), confirming the robustness of our methods. A detailed description of the quantification method was also added to the methods section.

Figure R1

14. In Fig 6D, what is the difference between + and ++? Please define them.

We apologize for the omission. In this figure, “-” signifies decreased staining compared to control cells; “+” means equal staining with controls, and “++” signifies stronger staining than controls (based on the quantification in Figure 7b). This clarification has been added to the figure legend.

15. Make notation consistent between the text and the figures. For example, Cep83 vs CEP83. Also check the notation of siRNA. I think notation like “siCEP120” is more common compared to “Si CEP120” as in the figures. This is also the case for “sgRNA”.

We have corrected each of these notations.

16. Typos. Page 9, 2nd line from the bottom; though -> through.
Page 13, last 2 sentences; check the articles and the tense (“the DC” and “co-localized”).

We apologize for these typos, and each of these errors has been corrected.

We thank this Reviewer for very helpful comments that have substantially improved our presentation.

Reviewer #2

1- Talpid3 knockout cells are defective in the removal of CP110 from the mother centrioles in response to serum starvation. CP110 suppresses centriole elongation by capping the distal end of the centrioles [1, 2]. However, centrioles are abnormally elongated in Talpid3 knockouts. Can the authors explain the mechanism of centriole elongation in the presence of CP110?

Furthermore, in the previous publication investigating the role of Talpid3 using RNAi [3], the authors did not report any defects to the centriole length which contradicts the present data. This need to be clarified.

First, we thank the Reviewer for his/her generally enthusiastic and very constructive review.

The Reviewer raises an interesting question about the persistence of CP110 in the face of abnormal centriole elongation. Several recent studies have suggested that there are other mechanisms that regulate centriole elongation without the requirement for

CP110 removal. For example, loss of OFD1⁸, over-expression of CPAP⁹, and unknown mechanisms in cancer cells¹⁰ have been published. We originally included the EM data showing aberrant centriole length in the *Talpid3* KO cells to indicate consistency with our previous siRNA experiments reported in Kobayashi et al. (ref. 11). We agree that centriole length control mediated by *Talpid3*, *C2CD3*, and *OFD1* is very interesting. However, the exact mechanism needs further study. Due to the primary focus of this paper, space limitations, and the length of time allotted to revising this manuscript, we propose to investigate this issue in future studies outside the scope of this manuscript.

Lastly, we respectfully point out that the defect in centriole length was indeed documented in Supplementary Figure 5 of a previous study using siRNA-mediated knock-down of *Talpid3*¹¹ and a recent report from another group¹².

2- The first section of the results is concluded by the statement that “the assembly of distal appendages is sufficient for recruitment of the IFT machinery but not for initiation of vesicle docking or subsequent events during ciliogenesis”. However, in the following paragraph they state that “all subsequent defects observed in *Talpid3*^{-/-} cells, including the failure to dock vesicles, TTBK2 recruitment, CP110 removal, and IFT transport could be attributed to the DA assembly defect”, which apparently contradicts their interpretation of their results.

We apologize for the lack of clarity and thank the Reviewer for pointing this out. Actually, both statements are correct, though in different contexts. The first statement is correct because it pertains to reintroduction of *Talpid3* fragments (spanning residues 400-700 or 466-1000) into *Talpid3*^{-/-} cells, which can rescue assembly of distal appendages and IFT88 localization but not vesicle docking, CP110 removal, or ciliogenesis (Fig. 2). The second statement refers to *Talpid3*^{-/-} cells without rescue by *Talpid3* fragments, and in this case, DA are completely absent, leading to all subsequent defects observed in *Talpid3*^{-/-} cells, including the failure to dock vesicles, TTBK2 recruitment, CP110 removal, and IFT transport. Nevertheless, we have modified our description accordingly to help clarify this point.

3- The authors attribute functions of several regions of *Talpid3* by generating truncation constructs. It would be important to describe the known domains of *Talpid3* and explain why they chose these fragments. Furthermore, I suggest using the word region or fragment instead of “domain” since the known domains of *Talpid3* are not described in the manuscript.

Thus far, only the coiled-coil domain has been identified as a domain as such in *Talpid3*, as shown in Figure 2. We chose all other fragments in an unbiased way with the aim of identifying functional regions. Accordingly, we have changed the word “domain” to “region” or “fragment” throughout the manuscript, except when referring to the coiled-coil domain.

4- On page 7, the authors report that they identified a region encompassing amino acids 466-700 of *Talpid3* which is required for asymmetric localization of Centrobin and Cep120. However, the data for this statement does not appear to be provided in Figure 1D.

The conclusion is based on the fact that fragments spanning residues 400-700 and 466-1000 can rescue centriole asymmetry as shown in original figure 1D (new Fig. 2). Although we did not generate the 466-700 fragment, our results imply that residues 400-465 are dispensable for this function. We have now clarified this conclusion on p.7.

5- The authors perform rescue experiments to identify the minimal regions of Talpid3 responsible for the functions reported in the manuscript. However, the major phenotype of regulating the asymmetric localization of daughter centriole specific-proteins, namely CEP120 and Centrobin, is not investigated and should.

We thank the Reviewer for this important suggestion, and we have now thoroughly investigated these proteins by immunofluorescence. Please see our newly added data in updated Fig. 2a and Fig. S1b.

6- On page 10, the authors claim that the centriolar localization of Talpid3 is essential for its interaction with CEP120 (sup. figure 1). However, in this figure C.1697A>T mutant also interacts with Flag-Cep120, albeit much weaker than 466-1000aa fragment.

We agree and have modified our statement accordingly.

7- The authors had previously reported the accumulation of centriolar satellite proteins PCM1, BBS4 and CEP290 in Talpid3-depleted cells [3]. However, to examine the role of centriolar satellites centriole maturation in the absence of Talpid3, they used a PCM1 knockout cell line which completely lacks centriolar satellites. It may be more relevant to investigate the role of centriolar satellites in centriole maturation using a cell line which displays the accumulation of centriolar satellites, similar to Talpid3^{-/-} cells.

We appreciate the suggestion raised by the Reviewer. We have taken two strategies to address the role of centriolar satellites (CS) in centriole maturation. First, we showed that PCM1 knock-out cells, which completely lack CS, are able to undergo normal daughter-to-mother centriole maturation (now Figs. 4a and 6b). Second, in C2CD3^{-/-} cells--wherein centriole maturation is defective, as it is in Talpid3^{-/-} cells--we observed a *decrease* in CS instead of an accumulation (new Fig.7b). We therefore conclude that defects in CS (accumulation observed in Talpid3^{-/-} cells or decreases in PCM1^{-/-} or C2CD3^{-/-} cells) are not likely to account for the blockage in centriole maturation. These conclusions have now been added to the manuscript (p.13).

8- In Figure 3E, the results of the transfection of N-terminal region of Talpid3 alone should be included. Moreover, the statement that “to some extent, the centriolar recruitment of OFD1 by Talpid3 plays an important role in the regulation of asymmetric localization of daughter centriole proteins” is not totally correct, since they showed that OFD1 appears dispensable for this phenomenon (Figure 3A&B).

We have included the results (obtained in triplicate) for the transfection of the N-terminal region of Talpid3 in Figure 1D. So to avoid redundancy (and further avoid adding to an already data-rich figure), we did not include it again in Fig. 3E (now Figure 4e). We agree that the above statement is not wholly accurate and have corrected it (p. 12).

9- In Figure 3E, the authors should present the results of transfecting cells with the N-terminal domain of Talpid 3 only, not fused to other proteins.

Please see point 8 above.

Minor points:

1- On page 6, the data on the accumulation of centriolar satellites in the absence of Talpid3 is presented in a previous publication, not in figure 1A. Please clarify this in the manuscript.

We apologize for this oversight. We have corrected this on page 6 and present and discuss the data in Fig. 7b and the corresponding section.

2- Please provide the quantification of the data presented in supp. Figure 1 for centriole duplication.

We now provide the quantification in new Fig. S1A.

3- FACS data in supp. Figure 2B. is pertaining to 3 cell lines including Talpid3 KO. Please include this in the figure legend.

We have modified the figure legend accordingly in the new Fig. S4b.

We are grateful to this Reviewer for a thorough critique that has substantively improved our manuscript.

References:

1. Schmidt, T.I., Kleylein-Sohn, J., Westendorf, J., Le Clech, M., Lavoie, S.B., Stierhof, Y.D., and Nigg, E.A. (2009). Control of centriole length by CPAP and CP110. *Curr Biol* 19, 1005-1011.
2. Spektor, A., Tsang, W.Y., Khoo, D., and Dynlacht, B.D. (2007). Cep97 and CP110 Suppress a Cilia Assembly Program. *Cell* 130, 678-690.
3. Kobayashi, T., Kim, S., Lin, Y.-C., Inoue, T., and Dynlacht, B.D. (2014). The CP110-interacting proteins Talpid3 and Cep290 play overlapping and distinct roles in cilia assembly. *The Journal of Cell Biology* 204, 215.

Reviewer #3

I find that the author's conclusion, if confirmed, is very novel and significant, and will greatly advance our knowledge on centriole and cilia biogenesis. Given its potential impact, I think the authors should carefully address the following concerns, which I believe will strengthen the data for their exciting/provocative proposal while resolving some inconsistency.

We are grateful for the very enthusiastic and highly complimentary appraisal of this Reviewer. We especially appreciate the assessment that our study is "very novel and significant and will greatly advance our knowledge." Nevertheless, the Reviewer raised some concerns, which we addressed below with extensive experimentation.

a). DCP REMOVAL IS REQUIRED FOR DISTAL APPENDAGE (DA) ASSEMBLY (Fig. 2 & 4): The authors should use 3D-SIM to confirm the rescue of DA assembly in both Talpid3^{-/-} and C2cd3^{-/-} cells by showing that these DA components are indeed assembled into a ring-like structure at the right location (distal end of the centriole) and size (~500 nm). Non-specific accumulation of DA proteins around the centrosome should not lead to such specific patterns.

We thank the Reviewer for this suggestion and agree that 3D-SIM data is important to support our conclusion. Importantly, to that end, we have now confirmed that knocking down Centrobin in Talpid3^{-/-} and C2CD3^{-/-} cells rescues DA assembly using 3D-SIM. Please see the images and quantification in new Fig. S2C. We would be happy to move

this figure to the main figures, at the discretion of the Reviewer and Editor, although we note that Fig. 2 (now Fig. 3) is already quite data-rich.

Similar experiments should be done for the *Ofd1* rescue data: Does centrobilin knockdown in *C2cd3*^{-/-} cells enable *Ofd1* to localize to the right location (centriole distal end) with the right pattern (small ring) or merely induce non-specific accumulation of *Ofd1* around the centrosome?

As in the above point, we confirmed that knocking down Centrobilin in *Talpid3*^{-/-} and *C2CD3*^{-/-} cells rescued centriolar OFD1 localization with 3D-SIM (please see images and quantitation in new Fig. S2C). Again, we would be happy to move this figure to the main figures, at the discretion of the Reviewer and Editor.

Similar experiments should be done for the CEP19 rescue data: true rescue or non-specific accumulation (Fig. 1D & 6A).

With 3D-SIM, we confirmed that reintroduction of a *Talpid3* fragment spanning residues 400-700 in *Talpid3*^{-/-} cells can rescue DA assembly and centrosomal CEP19 localization. Please see the images and quantification in new Fig. S1C.

b). *Ofd1* is known to be loaded to the distal end of pro-centrioles from the beginning of centriole biogenesis and stay there throughout the life of the centriole. *Ofd1* is also known to localize to the pericentriolar satellites, which normally accumulate around the centrosome/MTOC. In Fig 4, the authors reported that DCP removal is required for the recruitment of “centrosomal *Ofd1*”. What are these centrosomal *Ofd1* molecules? Are they those loaded during procentriole formation or those associating with the satellites or both? Does PACT-centrobilin or PACT-cep120 block the loading of *Ofd1* onto procentrioles, satellites or both (Fig. 4A)?

We agree that it is very important to clarify whether both the centrosomal and centriolar satellite (CS) pools of OFD1 participate in centriole maturation. Since our OFD1 antibody exclusively stained centrosomal OFD1⁸, we obtained a second OFD1 antibody (OFD1-2)¹³ to examine the CS pool of OFD1. Two new experiments and data support the conclusion that the centrosomal, but not the CS, pool of OFD1 is required for DA assembly. First, we show that ablation of PCM1 (using a PCM1 knock-out) disrupts the CS pool of OFD1, but, importantly, it does not affect DCP removal or DA assembly (please see new Supplementary Fig. 5a). Secondly, expression of PACT-Cep120 and PACT-Centrobilin blocks the centrosomal pool, but not the CS pool of OFD1 (Figure 5a and new Supplementary Fig. 5b). We have now added these results and corresponding discussion to the manuscript (page 13 and 21) and thank the Reviewer for this helpful suggestion.

c) I wonder if the authors know where in the centrosome PACT-centrobilin or PACT-cep120 is localized to, as both fusion constructs can associate with the PCM of the mother centriole via the PACT domain, and with the centriole (including procentrioles that do not have the PCM) via the domains residing in cep120 or centrobilin. In this sense, which population of PACT-centrobilin or PACT-cep120 inhibits the recruitment/loading of *Ofd1* (Fig. 4A)?

Previous studies showed that the PACT domain targets proteins to the PCM region close to the outer centriole wall¹⁴⁻¹⁶. An antibody against the PACT domain of PCNT stains a circle with a diameter of ~440 nm¹⁶, although we are not aware of high resolution studies to measure the diameter of PACT fusion proteins. We examined the localization of PACT-Cep120, PACT-Centrobilin, and PACT-Neur14 proteins by 3D-SIM and quantified the

diameter of the centrosomal ring formed by these fusion proteins (new Supplementary Fig. 5c). We measured the diameters of PACT fusion proteins as follows: 384 nm (PACT-120), 395 nm (PACT-Centrobins), and 388 nm (PACT-Neur14), suggesting that the PACT domain targets proteins to the PCM region close to the outer centriolar wall. Since endogenous DCPs do not localize symmetrically on both centrioles, the localization of PACT fusion proteins is determined, to a large degree, by the PACT domain. Also, the diameter of PACT fusion proteins is comparable to endogenous Cep120 (378 nm) and Centrobins (355 nm) rings on the DC (Fig. 6a), which were shown to localize to the region close to outer centriole wall^{1,2,16}. All of these data suggest that Cep120 and Centrobins in the PCM region close to the outer centriole wall inhibit the recruitment/loading of Ofd1. We have now added these results to the manuscript (page 15).

d). Other issues regarding localization studies: In some images, the authors used GT335 to mark individual centrioles (e.g. Fig. 3C), but in others (e.g. Fig. 2D), gamma-tubulin was chosen to mark individual centrosomes each of which may carry one (G1) or two (S/G2) centrioles. The interpretation of the localization data will be very different in each case. To fully reveal the nature of the sequential recruitment and removal of different DCPs in WT or mutant cells, the authors must use centriolar marker such as centrin or GT335 (instead of gamma-tubulin) to separately visualize G1 and S/G2 centrioles. Otherwise, I have hard time to understand some of the data presented here, especially the data for Ofd1 loading (see (b)).

The data in original Figs. 3C and 2D were obtained from cells after 24 hours of serum starvation. Therefore, the cells are at G0/G1 phase with two centriolar dots. So, we used GT335 and gamma-tubulin, which stained both centrioles, to check the localization of relevant proteins at G0/G1 phase. For the localization of DCPs and OFD1 at S/G2 phase in control and Talpid3^{-/-} cells, we used centrin to visualize the old and newly formed centrioles. Please see Supplementary Fig. 1a. These data suggest that DCPs cannot be removed, and OFD1 cannot be recruited to newly formed centrioles, in Talpid3^{-/-} cells in S/G2 phase. For the final comment regarding OFD1 loading (in point b), please see our answer above, which directly addresses this question.

e). From the EM study shown in Fig 1B, about 40% of centrioles in Talpid3^{-/-} cells are significantly elongated (> 1um). However, of all the IF images collected from Talpid3^{-/-} cells (e.g. Fig. 1A & E, Fig. 2B, Fig. 3C), none of the centrioles shown appear to be elongated (by GT335 staining). Why is that?

Conventional microscopy does not provide sufficient resolution to observe the centriole length defect, although in some cases, slightly elongated centrioles are nevertheless apparent (e.g. Fig. 1A GT335/Centrobins and Supplementary Fig. 1A Centrin/Neur14). EM and 3D-SIM provides substantially higher resolution and show the centriole elongation defect (Figure 1B and new Figure 6B).

f) Talpid3 and C2cd3 can regulate the property of the centriole distal end and centriolar satellites. In the deletion analysis shown in Fig. 1D, can the author address which activity of Talpid3 and C2cd3 is responsible for DCP removal? How can the authors be sure that it is the distal end associated activity that regulates DCP removal?

The Reviewer has raised an important point. There are two pools of C2CD3, the centriolar satellite (CS) pool and the centrosomal pool. To address this question, we have now examined the localization of centriolar and CS pools of C2CD3 by expressing myc-C2CD3 in PCM1^{-/-} cells (new Supplementary Fig. 5a), wherein centriole maturation is normal, but

centriolar satellites are not detectable (new Figs. 4a and 6b). We found that the centrosomal, but not the CS, pool of C2CD3 persists in *PCM1^{-/-}* cells. These data, together with other experiments in which we expressed PACT-DCP fusions (now Fig. 3a) or knocked down centrobilin (now Fig. 3b), suggest that the centrosomal, but not the CS, pool of C2CD3 is required for centriole maturation. We have never detected endogenous or ectopically expressed Flag-Talpid3 at CS in RPE1 cells^{11,12,17}, suggesting that the interaction between Talpid3 and proteins also found at the CS occurs primarily on centrioles. We conclude that Talpid3 function is restricted to the distal ends of centrioles rather than CS. We have now added these results and corresponding discussion to the manuscript (page 13 and 21) and thank the Reviewer for this helpful suggestion.

g). Personally, I find this paper not very easy to read. The amount of data is huge, supporting a very long story that is divided into parts not intuitively connected, lacking a primary focus. It will probably work better if the authors choose one primary finding as the focus and go deeper to reveal its significance. For example, for me, the connection between Talpid3/C2cd3/Ofd1 (centriolar satellites), DCP removal and DA assembly is very significant. I would love to see more detailed, focused analyses on this part only.

Our paper has primarily focused on the maturation of centriolar distal structures. To reiterate, we have, for the first time, identified the Talpid3/C2CD3/OFD1 complex at centriolar distal ends as required for proper removal of DCPs, assembly of distal and sub-distal appendages, centriolar satellites, actin network and the localization of CEP350/FOP/CEP19 complex. As stated by this Reviewer above, our work represents a major advance in the areas of centriole and cilia biogenesis. We have investigated how Talpid3 and C2cd3 regulate the asymmetry/removal of DCPs. Mechanistically, we found that it is not due to the interaction between Talpid3 and Cep120 (Supplementary Fig. 3) and found that the removal of DCPs is not due to protein degradation (Supplementary Fig. 4b) as inhibition of protein degradation by MG132 treatment does not affect DCPs removal (data not shown). We also examined the localization of PLK1 and PLK4, which has been implicated in centriole maturation, and we found that it is normal in *Talpid3^{-/-}* cells (data not shown). We are still working on other possible mechanisms to explain the links between daughter centriole protein disappearance and maturation, but it will clearly involve substantially more work beyond the scope of this study (and the timeframe allotted for revision).

We thank the Reviewer for helpful suggestions that have greatly strengthened our manuscript.

References

- 1 Mahjoub, M. R., Xie, Z. & Stearns, T. Cep120 is asymmetrically localized to the daughter centriole and is essential for centriole assembly. *The Journal of cell biology* **191**, 331-346, doi:10.1083/jcb.201003009 (2010).
- 2 Zou, C. *et al.* Centrobilin: a novel daughter centriole-associated protein that is required for centriole duplication. *The Journal of cell biology* **171**, 437-445, doi:10.1083/jcb.200506185 (2005).
- 3 Li, J. *et al.* Neurl4, a novel daughter centriole protein, prevents formation of ectopic microtubule organizing centres. *EMBO reports* **13**, 547-553, doi:10.1038/embo.2012.40 (2012).

- 4 Comartin, D. *et al.* CEP120 and SPICE1 cooperate with CPAP in centriole elongation. *Current biology : CB* **23**, 1360-1366, doi:10.1016/j.cub.2013.06.002 (2013).
- 5 Kurtulmus, B. *et al.* WDR8 is a centriolar satellite and centriole-associated protein that promotes ciliary vesicle docking during ciliogenesis. *Journal of cell science* **129**, 621-636, doi:10.1242/jcs.179713 (2016).
- 6 Conkar, D. *et al.* The centriolar satellite protein CCDC66 interacts with CEP290 and functions in cilium formation and trafficking. *Journal of cell science* **130**, 1450-1462, doi:10.1242/jcs.196832 (2017).
- 7 Ott, C. *et al.* VPS4 is a dynamic component of the centrosome that regulates centrosome localization of gamma-tubulin, centriolar satellite stability and ciliogenesis. *Scientific reports* **8**, 3353, doi:10.1038/s41598-018-21491-x (2018).
- 8 Singla, V., Romaguera-Ros, M., Garcia-Verdugo, J. M. & Reiter, J. F. Odf1, a human disease gene, regulates the length and distal structure of centrioles. *Developmental cell* **18**, 410-424, doi:10.1016/j.devcel.2009.12.022 (2010).
- 9 Schmidt, T. I. *et al.* Control of centriole length by CPAP and CP110. *Current biology : CB* **19**, 1005-1011, doi:10.1016/j.cub.2009.05.016 (2009).
- 10 Marteil, G. *et al.* Over-elongation of centrioles in cancer promotes centriole amplification and chromosome missegregation. *Nature communications* **9**, 1258, doi:10.1038/s41467-018-03641-x (2018).
- 11 Kobayashi, T., Kim, S., Lin, Y. C., Inoue, T. & Dynlacht, B. D. The CP110-interacting proteins Talpid3 and Cep290 play overlapping and distinct roles in cilia assembly. *The Journal of cell biology* **204**, 215-229, doi:10.1083/jcb.201304153 (2014).
- 12 Stephen, L. A. *et al.* TALPID3 controls centrosome and cell polarity and the human ortholog KIAA0586 is mutated in Joubert syndrome (JBTS23). *eLife* **4**, doi:10.7554/eLife.08077 (2015).
- 13 Lopes, C. A. *et al.* Centriolar satellites are assembly points for proteins implicated in human ciliopathies, including oral-facial-digital syndrome 1. *Journal of cell science* **124**, 600-612, doi:10.1242/jcs.077156 (2011).
- 14 Fu, J. & Glover, D. M. Structured illumination of the interface between centriole and peri-centriolar material. *Open biology* **2**, 120104, doi:10.1098/rsob.120104 (2012).
- 15 Lerit, D. A. *et al.* Interphase centrosome organization by the PLP-Cnn scaffold is required for centrosome function. *The Journal of cell biology* **210**, 79-97, doi:10.1083/jcb.201503117 (2015).
- 16 Lawo, S., Hasegan, M., Gupta, G. D. & Pelletier, L. Subdiffraction imaging of centrosomes reveals higher-order organizational features of pericentriolar material. *Nature cell biology* **14**, 1148-1158, doi:10.1038/ncb2591 (2012).
- 17 Wu, C. *et al.* Talpid3-binding centrosomal protein Cep120 is required for centriole duplication and proliferation of cerebellar granule neuron progenitors. *PloS one* **9**, e107943, doi:10.1371/journal.pone.0107943 (2014).

REVIEWERS' COMMENTS:

Reviewer #1 (Remarks to the Author):

The authors have addressed all of my comments. Although some issues were not clarified well for technical reasons, this reviewer is now supportive for publication of the revised version in Nature Communications.

Reviewer #2 (Remarks to the Author):

I have now read the revised version of the manuscript. The authors have done a good job addressing my concerns and I support publication of this work in Nat Comm.

Reviewer #2's comments on Reviewer #3's concerns:

Reviewer #2 only left confidential comments to the editor but considers that most of the requested experiments and clarifications have been addressed.